# When Can Proxies Improve the Sample Complexity of Preference Learning?

Yuchen Zhu [1]   Daniel Augusto de Souza [1]   Zhengyan Shi [1]   Mengyue Yang [2]   Pasquale Minervini [3]
Matt J. Kusner [4 5]   Alexander D'Amour [6]

## Abstract

We address the problem of *reward hacking*, where maximising a proxy reward does not necessarily increase the true reward. This is a key concern for Large Language Models (LLMs), as they are often fine-tuned on human preferences that may not accurately reflect a true objective. Existing work uses various tricks such as regularisation, tweaks to the reward model, and reward hacking detectors, to limit the influence that such proxy preferences have on a model. Luckily, in many contexts such as medicine, education, and law, a sparse amount of expert data is often available. In these cases, it is often unclear whether the addition of proxy data can improve policy learning. We outline a set of sufficient conditions on proxy feedback that, if satisfied, indicate that proxy data can provably improve the sample complexity of learning the ground truth policy. These conditions can inform the data collection process for specific tasks. The result implies a parameterisation for LLMs that achieves this improved sample complexity. We detail how one can adapt existing architectures to yield this improved sample complexity.

## 1. Introduction

Large Language Models (LLMs) have revolutionised machine learning with their surprising capabilities, surpassing human-level performance in law, medicine, and other examinations (Achiam et al., 2023; Amin et al., 2023). A large part of their success is their ability to incorporate human preferences to learn complex objectives such as trustworthiness (Yu et al., 2024), sentiment preferences (Chakraborty et al., 2024), and value alignment (Ji et al., 2023).

In many cases, this preference data is a *proxy* for the ground truth. For example, humans raters tend to prefer longer answers to a question, even if the answer is less informative (Zhou et al., 2024). In this case, 'response length' is a *proxy* for the true helpfulness of an answer. If an LLM is trained on this proxy data alone it leads to a 'length-bias' (Shen et al., 2023; Singhal et al., 2023), as LLMs fine-tuned with this preference data generate longer and better formatted responses to appear more helpful (Chen et al., 2024). This is an example of the well-known phenomenon of *reward hacking*[1]: a model optimised to perform well with respect to a proxy reward function, performs poorly with respect to a ground truth reward function (Casper et al., 2023). Reward hacking is a fundamental problem in learning that has been observed in optimised circuits listening in on the oscillators of other computers when instead tasked to build their own (Bird & Layzell, 2002), universities rejecting the most qualified applicants to boost their ratings (Golden, 2001), and many other cases in game playing (Clark & Amodei, 2016), autonomous driving (Knox et al., 2023), and text summarisation (Paulus et al., 2018).

To address reward hacking in LLMs, prior work largely designs tweaks to the model, data, and optimization procedure. This includes regularisation towards an initial policy (Schulman et al., 2017; Rafailov et al., 2023; Huang et al., 2024), changing properties of the reward model (Gao et al., 2023; Coste et al., 2024), using soft labels (Zhu et al., 2024), adjusting optimization hyperparameters (Singhal et al., 2023), reward hacking detection mechanisms (Pan et al., 2022; Miao et al., 2024), and introducing additional tools specialised to counteract length bias (Chen et al., 2024). The reasoning behind this comes from the makeup of proxy data. We can think of proxy data as having two parts: (i) a *true* part that brings a policy closer to the ground truth policy during learning and (ii) a *false* part that moves it farther away. Prior work limits learning to reduce the impact that the false part has on the final model.

Without any further information on proxy preferences or the ground truth, we are restricted to methods such as these, i.e., methods that are blind to the true and false parts of proxy data, to reduce the impact of reward hack-

[1]Department of Computer Science, University College London [2]University of Bristol [3]University of Edinburgh [4]Polytechnique Montréal [5]Mila - Quebec AI Institute [6]Google Deepmind. Correspondence to: Yuchen Zhu <yuchen.zhu.18@ucl.ac.uk>.

*Proceedings of the 42$^{nd}$ International Conference on Machine Learning*, Vancouver, Canada. PMLR 267, 2025. Copyright 2025 by the author(s).

---

[1]This is also sometimes referred to as *reward over-optimisation*.

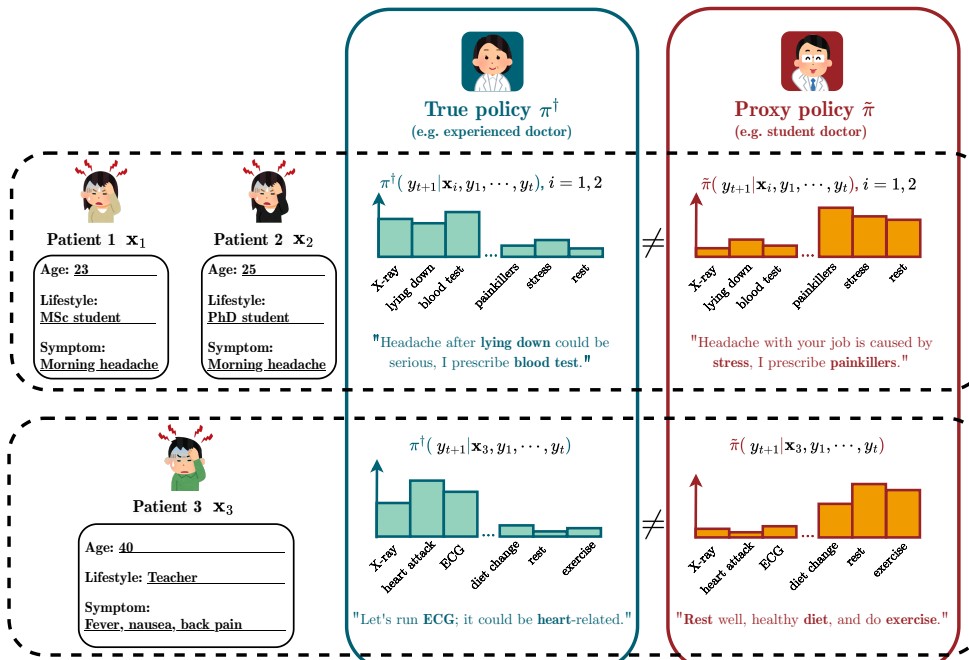

*Figure 1.* **A condition for useful proxy feedback.** (Illustrative purposes only. Not medical advice.) An expert (ground truth) doctor prescribes a different treatment compared with a student (proxy) doctor. However, Patients 1 and 2 are identified as lying *within the same level set of patients by both doctors*, based on their key characteristics - age, lifestyle and symptoms. Patient 3 has sufficiently different characteristics from Patient 1 and 2, and is identified to lie within a different level set by both doctors.

ing (Pan et al., 2022). Luckily, in many settings, we also have access to sparse observations of high-quality preferences (Daniels-Koch & Freedman, 2022). For instance, in demonstration-guided reinforcement learning, expert data is added to improve sample efficiency (Rajeswaran et al., 2017) and to guide exploration (Nair et al., 2018). Recent work has shown that including such expert information can help counter reward hacking in LLMs (Rita et al., 2024).

Consider the following medical example depicted in Figure 1. Patient 1 and Patient 2 consult the expert doctor and the student doctor about a condition they have. They have similar characteristics and essentially the same problem: a recurring morning headache that lasted a few days, but their exact phrasings can be different. Meanwhile, Patient 3 has different characteristics and a different condition to Patient 1 and 2. We think of the experienced doctor as representing a true policy and the student as a proxy policy. The two doctors both assign Patient 1 and 2 to the same group and Patient 3 to a different group, but the two doctors' recommendations for a given group are different, since the expert doctor can correctly recognised some easily misdiagnosed symptoms while the student doctor cannot.

We assume access to sparse prescriptions from an experienced doctor (ground truth) and plentiful prescriptions from less experienced student doctors (proxy). Even with ground truth data, If we naively learn a policy on the union of the dataset, we will learn a policy close to the proxy policy, as this data is more abundant. However, given the success of preference learning methods for LLMs, there is often

useful information to extract from the prolific proxy data. A natural question is: *When can proxy data ever provably improve preference learning?*

In this paper we outline a set of sufficient conditions on proxy feedback that, if satisfied, indicate that the proxy can provably improve the sample complexity of learning the ground truth policy. As not all proxies will satisfy these conditions, they can be used to guide a data collection process for a specific task. We show that as long as the collected proxy feedback shares certain properties with the true feedback, the sample complexity of learning with true preference data is provably improved by first training on large amounts of proxy preference data. The key idea behind this is that if the proxy and true policies share a certain structure, characterised in Condition 2-1, then it is possible to express the true policy as a low-dimensional adaptation of the proxy policy (Theorem 3). This relationship implies that certain parameters of the ground truth policy can be identified solely from proxy data, reducing the number of ground truth samples needed to learn the ground truth policy (Theorem 5, 6). This result immediately implies a parametrisation for LLMs that achieves improved sample complexity. Our contributions are:

- We characterise a set of sufficient conditions on proxy feedback such that the sample complexity of learning the true policy is reduced through learning on the proxy.

- We show that if proxy feedback satisfies the sufficient conditions, it implies a specific model parametrisation

and learning procedure to extract information from the proxies. We detail these and describe how one can adapt existing architectures to improve sample complexity.

## 2. Preliminaries

Consider the set of all prompts $\mathcal{X}$ and completions $\mathcal{Y}$, the elements of these sets are discrete sequences of tokens with arbitrary length, e.g. $\boldsymbol{x} = [x_1, \ldots, x_{N_{\boldsymbol{x}}}]$. By considering an enumeration of all completions $\mathcal{Y}$, the space of distributions $\mathcal{P}_{\mathcal{Y}}$ is equivalent to the subset of non-negative and unit norm sequences in the sequence space $\ell^1$. [2]

We think about a policy $\pi$ as a map from prompt space $\mathcal{X}$ to the *distribution* space $\mathcal{P}_{\mathcal{Y}}$. Starting from an initial policy $\pi_{\text{ref}}$[3], we want to find a target policy $\pi^{\dagger} : \mathcal{X} \to \mathcal{P}_{\mathcal{Y}}$ which aligns with the preferences of an ideal actor in a given scenario. To learn this policy, we have preference data directly from the ideal actor, denoted $\mathcal{D}^{\dagger}$, as well as preference data from a proxy actor, $\tilde{\mathcal{D}}$. The central question we consider is: under what assumptions can $\tilde{\mathcal{D}}$ improve the sample complexity of learning $\pi^{\dagger}$?

**Human preference feedback.** We aim to align $\pi_{\text{ref}}$ using preference data of the form $\{(\boldsymbol{x}, \boldsymbol{y}_w, \boldsymbol{y}_l)\}$, where $\boldsymbol{y}_w$ and $\boldsymbol{y}_l$ are candidate completions for prompt $\boldsymbol{x}$, and where $\boldsymbol{y}_w$ is preferred to $\boldsymbol{y}_l$. We assume these preferences are generated from a underlying scalar reward function $r(\boldsymbol{x}, \boldsymbol{y})$ according to Bradley & Terry (1952):

$$\boldsymbol{x} \sim p_{\mathcal{X}}; \quad \boldsymbol{y}_1, \boldsymbol{y}_2 \underset{\text{i.i.d.}}{\sim} \pi_{\text{ref}}(\cdot \mid \boldsymbol{x});$$

$$b \sim \text{Bern}[\sigma(r(\boldsymbol{x}, \boldsymbol{y}_1) - r(\boldsymbol{x}, \boldsymbol{y}_2))];$$

$$(\boldsymbol{y}_w, \boldsymbol{y}_l) = \begin{cases} (\boldsymbol{y}_1, \boldsymbol{y}_2) & \text{if } b = 1, \\ (\boldsymbol{y}_2, \boldsymbol{y}_1) & \text{if } b = 0. \end{cases},$$

where $\sigma(\cdot)$ is the sigmoid logistic function. We assume that $\boldsymbol{y}_1$ and $\boldsymbol{y}_2$ are sampled from $\pi_{\text{ref}}$ for simplicity, whereas in practice they can be sampled from other distributions over $\mathcal{P}_{\mathcal{Y}}$. In this model, higher relative rewards increase the chance of a completion being picked as the winner $\boldsymbol{y}_w$.

We assume that the true and proxy preference data, $\mathcal{D}^{\dagger}$ and $\tilde{\mathcal{D}}$, are generated by distinct reward functions $r^{\dagger}$ and $\tilde{r}$.

**Bandit problem setting.** Given a data-generating process $G = (r, \pi_{\text{ref}}, p_{\mathcal{X}})$ with reward function $r$, a reference policy $\pi_{\text{ref}}$, and a distribution of prompts $p_{\mathcal{X}}$, the optimal KL-regularised policy $\pi_G$ for the data generating process $G$ is

the one that maximises the following optimisation objective:

$$\arg\max_{\pi} \mathbb{E}_{\boldsymbol{x} \sim p_{\mathcal{X}}, \, \boldsymbol{y} \sim \pi(\cdot \mid \boldsymbol{x})}[r(\boldsymbol{x}, \boldsymbol{y})] - \beta \, \text{KL}(\pi(\boldsymbol{y} \mid \boldsymbol{x}) \parallel \pi_{\text{ref}}(\boldsymbol{y} \mid \boldsymbol{x})),$$

(1)

where the regularisation parameter $\beta$ controls how close to the reference the optimum should be. Under this objective, the optimal policy is given by:

$$\pi_G(\boldsymbol{y} \mid \boldsymbol{x}) \propto \pi_{\text{ref}}(\boldsymbol{y} \mid \boldsymbol{x}) \exp\left(\frac{1}{\beta} r(\boldsymbol{x}, \boldsymbol{y})\right). \quad (2)$$

The target policy we aim to learn is thus denoted $\pi^{\dagger}$ satisfying (2) with respect to $G^{\dagger} = (r^{\dagger}, \pi_{\text{ref}}, \mathcal{P}_{\mathcal{X}})$.

**Direct preference optimisation (DPO) and implicit rewards.** Following Rafailov et al. (2023), by optimising the following objective:

$$\arg\max_{\pi} \mathbb{E}_{(\boldsymbol{x}, \boldsymbol{y}_w, \boldsymbol{y}_l) \sim G}\left[\log \sigma\left(\beta \log \frac{\pi(\boldsymbol{y}_w \mid \boldsymbol{x})}{\pi_{\text{ref}}(\boldsymbol{y}_w \mid \boldsymbol{x})} - \beta \log \frac{\pi(\boldsymbol{y}_l \mid \boldsymbol{x})}{\pi_{\text{ref}}(\boldsymbol{y}_l \mid \boldsymbol{x})}\right)\right],$$

(3)

we recover the same optimal policy as described in Equation 2 without directly using any reward function. Thus, DPO avoids the need for reward modelling.

We note that the policy implicitly defines a reward function,

$$r(\boldsymbol{x}, \boldsymbol{y}) = \beta \log \frac{\pi(\boldsymbol{y} \mid \boldsymbol{x})}{\pi_{\text{ref}}(\boldsymbol{y} \mid \boldsymbol{x})}.$$

We can therefore define $\pi^{\dagger}, \tilde{\pi}$ as the policies that (implicitly) define $r^{\dagger}$ and $\tilde{r}$, respectively.

**True and proxy preference data** For many interesting tasks, sampling a dataset $\mathcal{D}^{\dagger}{}_{n^{\dagger}} := \{(\boldsymbol{x}_i, \boldsymbol{y}_{w,i}, \boldsymbol{y}_{l,i})\}_{i=1}^{n^{\dagger}}$ from $G^{\dagger} = (r^{\dagger}, \pi_{\text{ref}}, p_{\mathcal{X}})$ can be costly; therefore, the size $n^{\dagger}$ of the dataset might not be large enough for adequate training. In these cases, it is common to use a larger proxy dataset $\tilde{\mathcal{D}}_{\tilde{n}} := \{(\tilde{\boldsymbol{x}}_i, \tilde{\boldsymbol{y}}_{w,i}, \tilde{\boldsymbol{y}}_{l,i})\}_{i=1}^{\tilde{n}}$, where each data point is sampled i.i.d. from a proxy data-generating distribution $\tilde{G} = (\tilde{r}, \pi_{\text{ref}}, p_{\mathcal{X}})$, where $\tilde{G}$ and $G^{\dagger}$ only differ in the reward.

Nonetheless, if we do not have data from $G^{\dagger}$, then even if we have access to infinitely many data samples from $\tilde{G}$, and even if it allows us to learn the perfect reward model $\tilde{r}$[4], we can at best only learn the optimal *proxy* policy $\tilde{\pi}$, which differs from the optimal *true* policy in $\pi^{\dagger}$, *by construction* due to the difference in rewards.

## 3. Sufficient Conditions for Proxy Feedback

Theory and survey papers point out the difficulty of alignment under mismatch between the true reward function and

---

[2]$\ell^1$ is the space of sequences $\{y_i\}_i$ such that $\sum_{i=1}^{\infty} |y_i| < \infty$. The subset we consider is the subset of such sequences where all $y_i$ are non-negative and $\sum_{i=1}^{\infty} |y_i| = 1$.

[3]In practice, $\pi_{\text{ref}}$ is obtained from the supervised finetuning stage of language model training (Rafailov et al., 2023).

[4]The technical condition for this to be possible is to be provided $\mathcal{P}_{\mathcal{X}}$ and $\pi_{\text{ref}}$ have full support.

the one reflected by the human labelers (Skalse et al., 2022; Casper et al., 2023). In other related fields such as vision, (Chi et al., 2022) shows the impossibility of leveraging pre-training data for unseen tasks unless some similarity between the two tasks is given.

From these observations we can draw two conclusions. (i) We must have at least *some* data from $G^\dagger$ to learn $\pi^\dagger$; this motivates the need for both $\mathcal{D}_n^\dagger$ and $\hat{\mathcal{D}}_{\tilde{n}}$. (ii) In order for $\tilde{\pi}$ to inform us something about $\pi^\dagger$, they must share some similarities. Thus, we ask the following research question:

*What conditions allow for improved $G^\dagger$-sample efficiency in learning $\pi^\dagger$ after first training on $\tilde{\mathcal{D}}$? By how much?*

With the following conditions, we show that, when we have access to large amounts of proxy data, $\pi^\dagger$ can be expressed as a low-dimension adaptation of $\tilde{\pi}$, and hence the sample complexity of learning $\pi^\dagger$ is drastically improved, superexponential in the data manifold dimension.

In practice, these conditions can be helpful in guiding the design of the proxy data collection procedure. In particular, the conditions correspond to the expertise we might require proxy raters and true raters to share.

Our first technical condition concerns the similarity between the policy functions $\pi^\dagger$ and $\tilde{\pi}$.

**Condition 1.** *For a given metric $d_{\mathcal{P}_\mathcal{Y}}$ on the space of distributions on $\mathcal{Y}$, there is some positive scaler $L$ such that $d_{\mathcal{P}_\mathcal{Y}}(\pi^\dagger(\cdot|x_1), \pi^\dagger(\cdot|x_2)) \leq L d_{\mathcal{P}_\mathcal{Y}}(\tilde{\pi}(\cdot|x_1), \tilde{\pi}(\cdot|x_2))$.*

Informally, this says that if two prompts are mapped to very similar completion distributions by $\tilde{\pi}$, then they cannot get mapped to very different completion distributions by $\pi^\dagger$; for a unit difference of the former, the difference for the latter must not exceed $L$ for some positive scaler $L$.

This condition applies to situations where the proxy rater is within reasonable ballpark from the true rater: for example, for some medicines the correct dosage can vary by a large amount depending on the patient; if the expert doctor prescribes a certain dosage, and the student doctor prescribes a dosage different but close to that, then the condition can be considered satisfied. The condition also generalises to a broader situation: if the expert doctor's *change* in prescription, for instance, after observing some improvements in a patient, is similar to the *change* in prescription of a student doctor, then the condition can also be considered satisfied. However, in situations of crowdsourcing human preference from general populations, such as with Amazon Mechanical Turk, we cannot consider this condition to be satisfied; likewise, if we suspect that some proxy raters are adversarial, then we also cannot expect it satisfied.

Condition 1 implies another, easier-to-visualise, condition:

**Condition 2** (Shared level sets). *Given $x_1, x_2 \in \mathcal{X}$, we*

have: $\pi^\dagger(\cdot \mid x_1) = \pi^\dagger(\cdot \mid x_2)$ if $\tilde{\pi}(\cdot \mid x_1) = \tilde{\pi}(\cdot \mid x_2)$.

*Proof that Condition 1 implies Condition 2.* $\tilde{\pi}(\cdot \mid x_1) = \tilde{\pi}(\cdot \mid x_2)$ implies that $d_{\mathcal{P}_\mathcal{Y}}(\tilde{\pi}(\cdot|x_1), \tilde{\pi}(\cdot|x_2)) = 0$. Then notice $0 \geq d_{\mathcal{P}_\mathcal{Y}}(\pi^\dagger(\cdot|x_1), \pi^\dagger(\cdot|x_2)) \geq 0$, which then implies $\pi^\dagger(\cdot|x_1) = \pi^\dagger(\cdot|x_2)$. $\qquad\square$

Condition 2 says that two distinct prompts are mapped to the same response distribution under the true policy whenever they are under the proxy policy, and vice versa; mathematically, the two policies share level sets (Figure 2, left).

In the context of the running example (Figure 1), Condition 2 corresponds to the experienced doctor and the student doctor classifying the symptoms of patients in the same way. We could reasonably expect this because comprehending the relevant details of a patient's query is part of the basic training for a doctor. On the other hand, if the proxy preferences were sourced from generic crowd workers with no medical background, we would not expect this condition to be satisfied. Practically, survey instruments could be designed to directly assess this assumption, for example, by asking raters to rate the equivalence between different prompts.

Since $\tilde{\pi}$ and $\pi^\dagger$ share the same level sets, as supposed in Condition 2, it can be shown that $\pi^\dagger \circ \tilde{\pi}^{-1}|_{\tilde{\pi}(\mathcal{X})}$ is a well-defined function, where it should be noted that $\tilde{\pi}^{-1}|_{\tilde{\pi}(\mathcal{X})}(p)$ maps a point $p \in \tilde{\pi}(\mathcal{X})$ to its pre-image under $\tilde{\pi}$. The proof is provided in Appendix B.

**Lemma 1.** *Suppose Condition 2 is satisfied, $\pi^\dagger \circ \tilde{\pi}^{-1}|_{\tilde{\pi}(\mathcal{X})}$ is a well-defined function, where $\circ$ denotes function composition.*

Lemma 1 allows us to describe the 'difference' between the proxy and true policy as a function: given a distribution of completions $p$, $\pi^\dagger \circ \tilde{\pi}^{-1}|_{\tilde{\pi}(\mathcal{X})}$ maps *all* input prompts which share the response distribution $p$ by $\tilde{\pi}$ to a distinct response distribution, say $p'$, assigned by $\pi^\dagger$. This justifies learning an 'adapter' function which reassigns $p$ to the correct value $p'$. Had Condition 2 not held, then attempting to learn a function which assigns $p$ to $p'$ no longer makes sense since there could be some values of $p$ for which the corresponding $p'$ aren't unique.

Our next condition says that the set of expert response distributions is contained within the set of proxy response distributions (Figure 2, middle):

**Condition 3** (Image inclusion). $\pi^\dagger(\mathcal{X}) \subseteq \tilde{\pi}(\mathcal{X})$.

In the context of the running example (Figure 1), Condition 3 says that the student doctor could, in principle, express any valid medical advice distribution, even if the student doctor may not know how to assign them to appropriate symptoms with high accuracy. This is again a reasonable assumption when the proxy feedback comes from a student

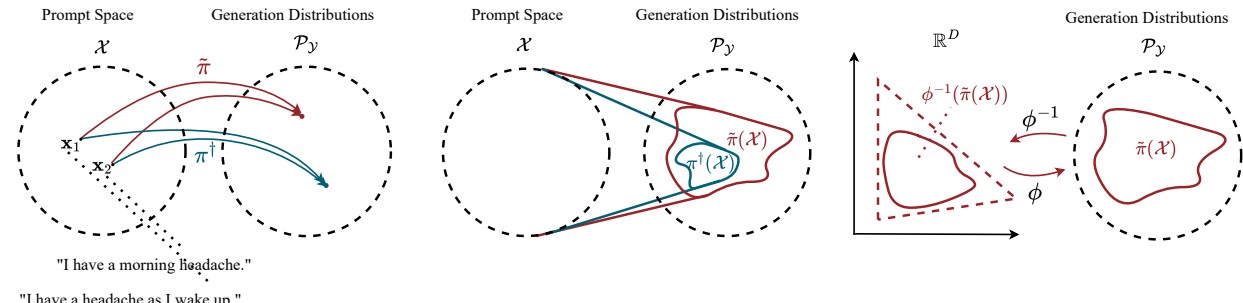

*Figure 2.* **Illustrations of Conditions 2-4**. Left, middle, right: Condition 2, 3, 4, respectively.

doctor, but is less plausible if the proxy feedback comes from a rater without general knowledge of core concepts that underlie medical advice. Practically, surveys could include questions to test the breadth of the rater's knowledge, or to gauge their educational background.

Finally, we need a technical condition similar to a common assumption underpinning many modern deep learning architectures: that data lie on a lower-dimensional manifold (Figure 2, right).

**Condition 4** (Finite-dimensional encoding of $\tilde{\pi}(\mathcal{X})$). *There exists an injective function $\phi\colon \mathcal{V} \to \mathcal{P}_{\mathcal{Y}}$, where $\mathcal{V} \subset \mathbb{R}^D$ is a bounded convex polytope with $D + 1$ vertices, such that:*

1. *The image $\phi(\mathcal{V})$ contains the image of the policies:*
   $\pi^{\dagger}(\mathcal{X}) \subseteq \tilde{\pi}(\mathcal{X}) \subseteq \phi(\mathcal{V})$;

2. *It is $(L_{\phi}, L_{\phi^{-1}})$-bi-Lipschitz with its left inverse $\phi^{-1}\colon \mathcal{P}_{\mathcal{Y}} \to \mathcal{V}$: $(1/L_{\phi^{-1}})\|v_1 - v_2\|_p \le d(\phi(v_1), \phi(v_2)) \le L_{\phi}\|v_1 - v_2\|_p$, where $d$ is a metric on $\mathcal{P}_{\mathcal{Y}}$, discussed later in Section 5.*

Note that the condition that $\mathcal{V}$ is a convex polytope is benign, since we can extend any bi-Lipschitz function which bijects a compact subset of $\mathbb{R}^N$ with $\tilde{\pi}(\mathcal{X})$ to a bi-Lipschitz function from a bounded convex polytope with the same Lipschitz constants to a set containing $\tilde{\pi}(\mathcal{X})$.

Condition 4 says that although the topological dimension of $\mathcal{P}_{\mathcal{Y}}$ can be extremely large, the image of $\tilde{\pi}$ is identified with a Euclidean subset only of dimension $D \ll \dim[\mathcal{P}_{\mathcal{Y}}]$. In particular, we provide a proof that the topological dimension of $\mathcal{P}_{\mathcal{Y}}$ can be as large as $\infty$; since any finite $n$-dimensional Euclidean space has topological dimension $n$, this shows that $\mathcal{P}_{\mathcal{Y}}$ is not a finite-dimensional Euclidean space.

**Proposition 2** (Topological dimension of $\mathcal{P}_{\mathcal{Y}}$ is $\infty$). *Let $\mathcal{K} = \{1, \cdots, k\}$ denote a set of $k$ tokens. Let $\mathcal{Y}$ be the set of all finite length token sequences with tokens from $\mathcal{K}$, and $\mathcal{P}_{\mathcal{Y}}$ the set of probability mass functions over $\mathcal{Y}$, then the topological dimension of $\mathcal{P}_{\mathcal{Y}} = \infty$. If $\mathcal{Y}$ is instead the set of token sequences with length $\le l$, then $\dim[\mathcal{P}_{\mathcal{Y}}] = O(k^l)$.*

*Proof.* Proof in Appendix A. □

In practical LLM training regimes, a maximum sequence length is implemented, but the dimension of $\mathcal{P}_{\mathcal{Y}}$ grows exponentially with $l$. In situations where the true and proxy policies generate responses from a small subset of all token sequences, it could be reasonable to expect Condition 4 to hold - for example, only a small subset of all token sequences form valid sentences, and an even smaller subset of those form valid medical advice, so we expect that for medical question-answering tasks the responses distributions can be encoded with fewer dimensions than for general question-answering tasks. The low-dimensional encoding can be viewed as some intermediate representation of the prompt that is sufficient for determining the response distribution. We can thus think of $\phi$ as a *policy decoder*.

## 4. Adapting $\tilde{\pi}$ to $\pi^{\dagger}$

Now, we derive an algorithm for learning $\pi^{\dagger}$ leveraging conditions 2-1. The algorithm hinges on a decompositon of the policy $\tilde{\pi}$ into an encoder $\tilde{\pi}$, a linear layer $\tilde{\Theta}$, and a decoder $\tilde{\phi}$, respectively. Importantly, we show that $\pi^{\dagger}$ can be expressed *reusing* these components from learning $\tilde{\pi}$ using $\tilde{\mathcal{D}}$, with the addition of a *low-dimensional adapter function between known spaces*. This reduction to learning an adapter function allows us to derive a drastic sample complexity improvement (Theorem 5).

We outline the main steps of the derivation now:

1. First, from Condition 3 and 4, we can show that both $\tilde{\pi}$ and $\pi^{\dagger}$ map prompts into a common lower-dimensional space before decoding into response distributions.

2. Then, by Condition 2 and 1, it can be shown that $\tilde{\pi}$ and $\pi^{\dagger}$ differ only by a Lipschitz continuous function mapping $\Delta^D \to \Delta^D$.

### 4.1. Factorising $\pi^{\dagger}$ and $\tilde{\pi}$ through $\mathcal{V}$.

**Notation.** In this section, we make heavy use of the mathematical concept of function composition; $f \circ g$ denotes '$f$

composes $g$', and is a function which first maps an input $x$ through $g$, and then maps $g(x)$ through $f$ to $f(g(x))$.

By Condition 4, the proxy policy $\tilde{\pi}$ factors through $\mathcal{V}$: that is, $\tilde{\pi}$ maps from the space of prompts to the space of response distributions via some intermediate representation of the prompt (i.e., $\mathcal{V}$) sufficient for determining the response distribution. Specifically, there is a bi-Lipschitz injective decoder $\phi$ from $\mathcal{V}$ to the image of the proxy policy $\tilde{\pi}(\mathcal{X})$. Additionally, Condition 3 says that the image of $\pi^\dagger$ is included in the image of $\tilde{\pi}$.

Therefore, we can view $\pi^\dagger$ as a function composition of some decoder $\phi : \mathcal{V} \to \tilde{\pi}(\mathcal{X})$[5] and some *encoder* function $\mathcal{X} \to \mathcal{V}$, such that

$$\pi^\dagger : \mathcal{X} \xrightarrow{\phi^{-1} \circ \pi^\dagger} \mathcal{V} \xrightarrow{\phi} \tilde{\pi}(\mathcal{X}) \tag{4}$$

Analogously, $\tilde{\pi}$ can be viewed also as

$$\tilde{\pi} : \mathcal{X} \xrightarrow{\phi^{-1} \circ \tilde{\pi}} \mathcal{V} \xrightarrow{\phi} \tilde{\pi}(\mathcal{X}) \tag{5}$$

Next, we show that $\pi^\dagger$ can be expressed by inserting a transformation into a function decomposition of $\tilde{\pi}$. This transformation can be shown to be a Lipschitz map between two known $D$-dimensional spaces. We can thus think of this transformation as an 'adapter' function.

### 4.2. $\pi^\dagger$ and $\tilde{\pi}$ differ by a function between $D$-simplices.

It is now possible to show that $\pi^\dagger$ and $\tilde{\pi}$ differ only by a transformation on the representation space $\mathcal{V}$. However, for the sample complexity arguments that follow, it is convenient to map $\mathcal{V}$ to a $D$-simplex $\Delta^D$, then to show that $\pi^\dagger$ and $\tilde{\pi}$ differ only by a transformation on $\Delta^D$.

To this end, since $\mathcal{V}$ is a $D$-polytope with $D + 1$ vertices, every point in $\mathcal{V}$ can be expressed as a convex combination of the vertices. Then, it can be shown that $\tilde{\pi}$ and $\pi^\dagger$ can be further factored through a $D$-simplex, $\Delta^D$. With this formalism in hand, we now state our result. The proof is provided in Appendix B.

**Theorem 3.** *We work under Conditions 2, 3, 4 and 1. For some D, there exists a Lipschitz invertible function $\tilde{\phi} : \mathcal{V} \to \tilde{\pi}(\mathcal{X})$ satisfying Condition 4, $\tilde{\Theta} \in \mathbb{R}^{N \times (D+1)}$ and $\tilde{\tau}^\circ : \mathcal{X} \to \Delta^D$ s.t. $\tilde{\pi} = \tilde{\phi} \circ \tilde{\Theta}\tilde{\tau}^\circ$.*

*Moreover, for any $\left( \tilde{\phi}, \tilde{\Theta}, \tilde{\tau}^\circ \right)$ such that $\tilde{\pi} = \tilde{\phi} \circ \tilde{\Theta}\tilde{\tau}^\circ$, there exists a Lipschitz continuous function $\bar{\pi}^\dagger : \Delta^D \to \Delta^D$ s.t. $\pi^\dagger = \tilde{\phi} \circ \tilde{\Theta}\bar{\pi}^\dagger \circ \tilde{\tau}^\circ$.*

Theorem 3 has two important implications for learning. 1. It establishes that there exists a decomposition of $\tilde{\pi}$ with

---

[5]Strictly, after $\phi^\dagger$ we still need to go through an inclusion to land in $\mathcal{P}_\mathcal{Y}$, but to simplify notation we omit this technicality.

modules that can be reused to express $\pi^\dagger$. 2. It further establishes that *for any* satisfactory decomposition, there exists an adapter $\bar{\pi}^\dagger$. This suggests that in practice we can first find a suitable triplet $\left( \tilde{\phi}, \tilde{\Theta}, \tilde{\tau}^\circ \right)$, then learn an adapter.

### 4.3. Model Parametrisation and Learning

We now sketch our learning algorithm. Theorem 3 gives rise to a two-step procedure to learn the true policy $\pi^\dagger$, firstly we recover the functional components using a large proxy dataset $\tilde{\mathcal{D}}_{\tilde{n}}$ and then, secondly, we use a small true dataset $\mathcal{D}^\dagger_{n^\dagger}$ to learn the low-dimensional adapter.

**Stage 1** Based on Theorem 3, we model the proxy policy $\tilde{\pi}$ with a parametric model composed of three functions: (i) $\tilde{\tau}^\circ_\theta$, an embedding function from the prompts $\mathcal{X}$ to the $D$-simplex $\Delta^D$, (ii) $\tilde{\Theta} \in \mathbb{R}^{D \times (D+1)}$, a linear map from the simplex to a convex polytope $\mathcal{V}$, and (iii) $\tilde{\phi}_\theta$, an injective function from the latent space $\mathcal{V}$ to a distribution of completions $\mathcal{P}_\mathcal{Y}$. Therefore, our model is expressed as:

$$\tilde{\pi}_\theta = \tilde{\phi}_\theta \circ \tilde{\Theta}\tilde{\tau}^\circ_\theta. \tag{6}$$

Based on the DPO loss (Eq. 3), $\tilde{\phi}_\theta$, $\tilde{\Theta}$, and $\tilde{\tau}^\circ_\theta$ are learned using the empirical preference learning objective with the proxy dataset $\tilde{\mathcal{D}}_{\tilde{n}}$:

$$\tilde{\mathrm{L}}_{\tilde{n}}(\tilde{\pi}_\theta) = -\frac{1}{\tilde{n}} \sum_{i=1}^{\tilde{n}} \log \sigma \left( \beta \log \frac{\tilde{\pi}_\theta(\tilde{\boldsymbol{y}}_{w,i} \mid \tilde{\boldsymbol{x}}_i)}{\pi_{\mathrm{ref}}(\tilde{\boldsymbol{y}}_{w,i} \mid \tilde{\boldsymbol{x}}_i)} - \beta \log \frac{\tilde{\pi}_\theta(\tilde{\boldsymbol{y}}_{l,i} \mid \tilde{\boldsymbol{x}}_i)}{\pi_{\mathrm{ref}}(\tilde{\boldsymbol{y}}_{l,i} \mid \tilde{\boldsymbol{x}}_i)} \right). \tag{7}$$

In the large sample limit, the optimal parametrised model $\tilde{\phi}^*_\theta \circ \tilde{\Theta}^* \tilde{\tau}^{\circ,*}_\theta$ minimises the population proxy preference loss, and due to Theorem 3, we know that the optimal KL-regularised proxy policy $\tilde{\pi} = \tilde{\phi}^*_\theta \circ \tilde{\Theta}^* \tilde{\tau}^{\circ,*}_\theta$, thus, justifying our parametrization.

**Stage 2** Following this, we then model the true policy $\pi^\dagger$ using the same pre-trained components from our model of the proxy policy $\tilde{\pi}_\theta$ with the addition of a low-dimensional adapter function $\bar{\pi}^\dagger_\theta$ which maps a latent representation in the simplex $\Delta^D$ to another $\Delta^D$ as follows:

$$\pi^\dagger_\theta = \tilde{\phi}_\theta \circ \tilde{\Theta}\bar{\pi}^\dagger_\theta \circ \tilde{\tau}^\circ_\theta. \tag{8}$$

The adapter $\bar{\pi}^\dagger_\theta$ is learned by optimization of the empirical preference learning objective with the true dataset $\mathcal{D}^\dagger_{n^\dagger}$, while keeping $\tilde{\phi}_\theta$, $\tilde{\Theta}$, and $\tilde{\tau}^\circ_\theta$ fixed based on their previously optimised values:

$$\mathrm{L}^\dagger_{n^\dagger}\left( \bar{\pi}^\dagger_\theta \right) = -\frac{1}{n^\dagger} \sum_{i=1}^{n^\dagger} \log \sigma \left( \beta \log \frac{\tilde{\phi}_\theta\left( \tilde{\Theta}\bar{\pi}^\dagger_\theta(\tilde{\tau}^\circ_\theta(\boldsymbol{x}_i)) \right)[\boldsymbol{y}_{w,i}]}{\pi_{\mathrm{ref}}(\boldsymbol{y}_{w,i} \mid \boldsymbol{x}_i)} - \beta \log \frac{\tilde{\phi}_\theta\left( \tilde{\Theta}\bar{\pi}^\dagger_\theta(\tilde{\tau}^\circ_\theta(\boldsymbol{x}_i)) \right)[\boldsymbol{y}_{l,i}]}{\pi_{\mathrm{ref}}(\boldsymbol{y}_{l,i} \mid \boldsymbol{x}_i)} \right). \tag{9}$$

There may be more efficient algorithms which learn the triplet $\left( \tilde{\phi}, \tilde{\Theta}, \tilde{\tau}^\circ \right)$ by using the proxy and true data simultaneously. Nonetheless, by splitting the learning into two

stages, the first using only proxy data, and the second using only true data, we can make a direct sample complexity comparison between learning $\pi^\dagger$ from scratch, and learning the adapter $\bar\pi^\dagger$ in Stage 2, in terms of the size of the true data $\mathcal{D}^\dagger_{n^\dagger}$.

# 5. Convergence Rates Analysis

To illustrate the benefit of learning $\pi^\dagger$ using the outlined algorithm, we analyse its soundness by showing the sample complexity improvement given that we have identified the true $\tilde\phi$, $\tilde\Theta$ and $\tilde\tau^\circ$ from the proxy dataset in the first stage. This can be a reasonable approximation of the properties of the learning procedure in cases where the proxy dataset is much larger than the true dataset. To this end, we analyse the generalisation error bound for the second stage given access to true $\tilde\phi$, $\tilde\Theta$ and $\tilde\tau^\circ$. Following the approaches of Elesedy (2022), Mohri et al. (2012, Exercise 3.31), the generalisation error can be shown to be linear in the covering number of the hypothesis class. Our insight here is that the hypothesis class of $\pi^\dagger$ is made smaller by having knowledge of $\tilde\phi$, $\tilde\Theta$ and $\tilde\tau^\circ$, hence the covering number is also smaller. In order to define covering numbers, we first define a notion of metric on all relevant spaces and the hypothesis classes we consider.

## 5.1. Metrics and Hypothesis Classes

**Metric on finite-dimensional spaces.** For any subset of the Euclidean space, we use the $p$-norm-induced metric; for a simplex $\Delta^D$ we denote its metric by $d_\Delta$ and for any other finite dimensional space $\mathcal{U}$ we use $d_\mathcal{U}$.

**Metric on the prompt space.** The prompt space $\mathcal{X}$, is a discrete and unstructured space, so we define a metric, $d_\mathcal{X}$ based on some fixed embedding function $f$, which maps a prompt to a vector space with finite but high dimensions: $d_\mathcal{X}(x, x') = d_{f(\mathcal{X})}(f(x), f(x')) = \|f(x) - f(x')\|_p$.

Intuitively, we can think about $f$ as some general-purpose embedding, such as one obtained by retrieving some intermediate layer from a large model such as CLIP (Radford et al., 2021). Importantly, this metric is only relevant when considering the complexity of the hypothesis class when we learn the target policy without proxy data, which one expects to be large; this intuition is confirmed since a general-purpose embedding may not work well across all tasks, and can result in a large Lipschitz constant for the target function.

**Metric on a policy space.** Defining a metric on a space of policies is more involved, and we begin by relating a policy and the reward function under which it is the KL-regularised optimal policy. Given a policy $\pi : \mathcal{X} \to \mathcal{P}_\mathcal{Y}$, define the

implicit reward as:

$$r_\pi(\boldsymbol{x}, \boldsymbol{y}) = \beta \log \frac{\pi(\boldsymbol{y} \mid \boldsymbol{x})}{\pi_{\text{ref}}(\boldsymbol{y} \mid \boldsymbol{x})} \tag{10}$$

Using $r_\pi$, we can define a metric on the set of policies, and we express in the next lemma. The proof is in Appendix C.

**Lemma 4** (Metric on policies through $r_\pi$). *For any policies $\pi, \pi' : \mathcal{X} \to \mathcal{P}_\mathcal{Y}$, so that $\sum_\mathcal{Y} \pi(y \mid x) = \sum_\mathcal{Y} \pi'(y \mid x) = 1$ for any $x \in \mathcal{X}$, define the function*

$$d_r(\pi, \pi') = \|r_\pi - r_{\pi'}\|_\infty \tag{11}$$

*Then, if the rewards are bounded, $\|r_\pi\|_\infty < \infty$ and $\|r_{\pi'}\|_\infty < \infty$, it is a metric in the space of policies.*

**Metric on the completion distribution space.** Additionally, we define a metric on the space of distributions of completions, $\mathcal{P}_\mathcal{Y}$. Motivated by the metric on policies through the rewards, we define the function $d_{\mathcal{P}_\mathcal{Y}} : \mathcal{P}_\mathcal{Y} \times \mathcal{P}_\mathcal{Y} \to \mathbb{R} \cup \{\infty\}$:

$$d_{\mathcal{P}_\mathcal{Y}}(p, q) = \left\| \beta \log \frac{p(\cdot_y)}{q(\cdot_y)} \right\|_\infty \tag{12}$$

Now define the hypothesis class of $\pi^\dagger$ as:

**Definition 1** (Hypothesis class $\mathring\Pi$). *Let $\mathring\Pi$ be a set of policies $\pi : \mathcal{X} \to \mathcal{P}_\mathcal{Y}$, so that, for all $x$, $\sum_y \pi(y|x) = 1$, and have reward at most $C$, meaning that $\|r_\pi\|_\infty \le C$.*

From Lemma 4, $\mathring\Pi$ and all its subsets are metric spaces when endowed with $d_r$. Later on, we will show that the covering number of a hypothesis class depends on the smallest Lipschitz constant of the class.

Meanwhile, based on the exitance of $\tilde\phi$, $\tilde\Theta$, $\tilde\tau^\circ$, we define a subset of $\mathring\Pi$:

**Definition 2** (Hypothesis class fixing $\tilde\phi$, $\tilde\Theta$, $\tilde\tau^\circ$ and Lipschitz–constant $L_{\bar\pi}$). *Fix $\tilde\phi$, $\tilde\Theta$, $\tilde\tau^\circ$. Let $\Pi\left(\tilde\phi, \tilde\Theta, \tilde\tau^\circ, L_{\bar\pi}\right) \subseteq \mathring\Pi$ the set containing all $\pi \in \mathring\Pi$ which can be written as*

$$\pi(\cdot|x) = \tilde\phi \circ \tilde\Theta\bar\pi \circ \tilde\tau^\circ(x) \tag{13}$$

*for some $L_{\bar\pi}$-Lipschitz $\bar\pi$, with respect to $d_\Delta$.*

Finally, we define the loss functions of our setup as

**Definition 3** (True and empirical DPO risks). *Following Rafailov et al. (2023), we define the true risk as*

$$R_{G^\dagger}(\pi) = \mathbb{E}_{G^\dagger} \left[ \log \sigma \left( \beta \log \frac{\pi(Y_w|X)}{\pi_{ref}(Y_w|X)} - \beta \log \frac{\pi(Y_l|X)}{\pi_{ref}(Y_l|X)} \right) \right] \tag{14}$$

*where $G^\dagger = (r^\dagger, \pi_{ref}, p_\mathcal{X})$ is the true data generating process.*

*The empirical risk is the Monte Carlo estimate of $R_{G^\dagger}(\pi)$,*

$$R_{\mathcal{D}^\dagger_{n^\dagger}}(\pi) = \frac{1}{n^\dagger} \sum_{i=1}^{n^\dagger} \left[ \log \sigma \left( \beta \log \frac{\pi(y_{w,i}|x_i)}{\pi_{ref}(y_{w,i}|x_i)} - \beta \log \frac{\pi(y_{l,i}|x_i)}{\pi_{ref}(y_{l,i}|x_i)} \right) \right] \tag{15}$$

## 5.2. Main Results

Now we are ready to state our main sample complexity result. The proof is given in Appendix C.1.

**Theorem 5** (Sample complexity of learning with proxy). *The covering number of the class of adapted policies $\Pi$ is bounded by a function of the latent space's dimension $D$:*

$$\text{Cov}\left(\Pi\left(\tilde{\phi}, \tilde{\Theta}, \tilde{\tau}^\circ, L_{\bar{\pi}}\right), d_r, 3\kappa + 3L_\phi \|\tilde{\Theta}\|_p L_{\bar{\pi}} \delta\right)$$
$$\leq \left(\frac{2L_\phi \|\tilde{\Theta}\|_p \sqrt{D}}{\kappa}\right)^{D\left(\frac{2\sqrt{D}}{\delta}\right)^D} \quad (16)$$

*where $\kappa$ is a free parameter. If we set $\kappa = \frac{\epsilon}{48}$, then we need*

$$n(\epsilon, \omega) = \Omega\left(\frac{D}{\epsilon^2}\left(\frac{96 L_\phi \|\tilde{\Theta}\|_p L_{\bar{\pi}} \sqrt{D}}{\epsilon}\right)^D \log\left(\frac{96 L_\phi \|\tilde{\Theta}\|_p L_{\bar{\pi}} \sqrt{D}}{\epsilon}\right) - \log \omega\right)$$

*samples to generalise. That is, whenever $n' \geq n(\epsilon, \omega)$,*

$$P\left(\sup_{\pi \in \Pi\left(\tilde{\phi}, \tilde{\Theta}, \tilde{\tau}^\circ, L_{\bar{\pi}}\right)} |R_{G^\dagger}(\pi) - R_{\mathcal{D}^\dagger_{n^\dagger}}(\pi)| \geq \epsilon\right) \leq \omega \quad (17)$$

*So, with probability at most $\omega$, the worse gap between the true and empirical risks is at least $\epsilon$.*

For comparison, we also state the sample complexity result for learning without adapting a proxy policy. The proof is in Appendix C.2.

**Theorem 6** (Sample complexity of learning without proxy). *Let $D'$ be the dimension of a given embedding function $f$ of $\mathcal{X}$, and let $\mathring{\Pi}\left(L_\phi \|\tilde{\Theta}\|_p L_{\bar{\pi}}\right)$ be the subset of $\mathring{\Pi}$ where $\pi$ is $L_\phi \|\tilde{\Theta}\|_p L_{\bar{\pi}}$-Lipschitz. We need*

$$n(\epsilon, \omega) = \Omega\left(\frac{D'}{\epsilon^2}\left(\frac{48 L_\phi \|\tilde{\Theta}\|_p L_{\bar{\pi}} E'(p, D') \sqrt{D'}}{\epsilon}\right)^{D'} \log\left(\frac{48 L_\phi \|\tilde{\Theta}\|_p L_{\bar{\pi}} E'(p, D') \sqrt{D'}}{\epsilon}\right) - \log \omega\right)$$

*samples to generalise, where $D' \gg D$, and $E'(p, D') \gg 1$. That is, whenever $n' \geq n(\epsilon, \omega)$, we have*

$$P\left(\sup_{\pi \in \mathring{\Pi}\left(L_\phi \|\tilde{\Theta}\|_p L_{\bar{\pi}}\right)} |R_{G^\dagger}(\pi) - R_{\mathcal{D}^\dagger_{n^\dagger}}(\pi)| \geq \epsilon\right) \leq \delta \quad (18)$$

*So, with probability at most $\omega$, the worse gap between the true and empirical risks is at least $\epsilon$.*

**Discussion.** Theorem 6 says that if we learn $\pi^\dagger$ directly from expensive samples of $G^\dagger$, then the sample complexity scales with $D'$, which is the dimension of the embedding space ; this can be extremely large. However, if we parametrise $\pi^\dagger$ using $\tilde{\phi}, \tilde{\Theta}, \tilde{\tau}^\circ$ which compose to be $\tilde{\pi}$, and can be learned from cheap samples of $\tilde{G}$, then Theorem 5 asserts that the number of *expensive* samples we need from $G^\dagger$ scale with the latent space dimension $D$ which can be much smaller than the embedding dimension $D'$.

**Experiments.** While this work focuses on theory and verifies the claims through mathematical proofs, we provide in Appendix D a small-scale experiment on over-smoothing in reward learning, as well as an empirical validation of Theorem 5 and Theorem 6.

# 6. Related Work

**Reward hacking theory.** Initial work on the theory of reward hacking considered the setting where the proxy reward was a function of a subset of true reward features (Zhuang & Hadfield-Menell, 2020). This work demonstrates that optimising the proxy can lead to arbitrarily low true reward. Similarly, Tien et al. (2022) give theoretical results for reward hacking when a learned reward uses nuisance variables that correlate with true causal variables. These ideas were extended to arbitrary MDPs by (Skalse et al., 2022) who define a proxy reward as hackable if it prefers policy $\pi_1$ over $\pi_2$ when the true reward has the opposite preference. Recent work has sought to develop scaling laws for reward hacking that describe how the true reward changes as the proxy reward is optimised (Gao et al., 2023). Rafailov et al. (2024) show similar over-optimisation patterns in DPO at higher KL-divergence budgets, even without an explicit reward model. In contrast to these works, our theoretical results suggest a new model parametrisation and training scheme which achieves improved sample complexity for learning the true policy; hence, our results are constructive.

**Addressing reward hacking in LLMs.** One of the classic examples of reward hacking in LLMs is their propensity for verbose responses that are not more helpful, often called the 'length bias', or 'length hacking' of LLMs (Singhal et al., 2023). To address this, Singhal et al. (2023) modified various aspects of PPO (increasing KL regularisation, omitting outputs beyond a certain length, and reward scaling), as well as the training data, with mixed success. Chen et al. (2024) conducted a large-scale evaluation of the impact of hyperparameters and the above modifications on reward hacking. They further introduce a model that decorrelates preference predictions with length. Miao et al. (2024) formulate reward modelling as optimising a variational information bottleneck and then use this to filter out less important features in latent space. Huang et al. (2024) mitigates reward over-optimisation by replacing the KL regularisation with an alternative term which implicitly implements the principle of pessimism in the face of uncertainty. Yang et al. (2024) addresses a form of reward misalignment due to distribution shift of the prompts and responses seen in training versus test time. To the best of our knowledge, reward hacking due to a difference in the reward functions in training and test time is not discussed explicitly in existing work. More importantly, the impossibility of target policy recovery without *some* data from the target reward, is not yet acknowledged.

**RLHF with expert feedback.** Human feedback often varies in quality, and one key challenge is how to incorporate these different feedbacks into learning (Daniels-Koch & Freedman, 2022). Freedman et al. (2023) formulate selecting which human to query for feedback as a bandit problem. Yamagata et al. (2024) uses the Boltzmann-rational model to account for varying levels of expertise. Our model parametrisation leverages certain invariances between proxy and expert/true feedback, allowing identification of the true policy as a low-dimensional adaptation of the proxy policy.

**Domain adaptation in the wider transfer learning and domain adaptation literature.** Reward hacking describes the general problem of adapting to a new domain from what's represented by training data, and appears in many sub-fields of transfer learning and domain adaptation. For instance, poisoning attacks in RL often consider data distributions polluted by an adversary, meaning that the data distribution represented by adversary is chosen in order to minimise rewards obtained by the targeted agent (Pinto et al., 2017), but this differs from our setup since we do not consider the adversarial aspect. Close to us is a class of subspace methods, where a common subspace between target and training environments is learned (Gopalan et al., 2011; Gong et al., 2012; Fernando et al., 2013), but this line of works do not analyse sample complexity of the target environment, and can be computationally expensive. A line of research directly considering how regrets scale with the number of samples seen considers multi-fidelity bandits. Analysis in this case assumes that the absolute difference in rewards between the low and high fidelity bandits is bounded by a known quantity (Kandasamy et al., 2016), whereas in our work the difference between the proxy and true rewards need not be bounded at all, let alone by a known quantity.

## 7. Conclusion

We study the problem of reward hacking due to distribution shifts. Specifically, an abundance of preference rankings is generated by a proxy reward function, different from the true reward function, which is costly to query. We thus consider the setting where we have a large proxy dataset and a small true dataset; to the best of our knowledge, we are the first to consider this setting. We formulate conditions motivated by a real-world example, under which we prove that the optimal proxy policy can be decomposed into component functions shared with the optimal true policy, and that the true policy is only one low-dimensional adapter function away from the proxy policy given the shared component functions. We then observe that in the large sample limit of proxy data, one set of such component functions can be identified from minimising the preference loss. Leveraging this, we provide a characterisation of the sample complexity bound for learning the hypothesis class both with and without

knowledge of the shared component functions; in particular, it is seen that under knowledge of the shared component functions the sample complexity bound is much lower than without such knowledge. In future work we aim to further investigate these findings empirically, as well as investigate relaxations of the conditions above.

ACKNOWLEDGMENTS

YZ acknowledges support by the Engineering and Physical Sciences Research Council with grant number EP/S021566/1.

## Impact Statement

Preference learning lies at the core of RLHF. Additionally, due to the costliness of high-quality data collection, proxy data is frequently used in practice. In this setting, our work clarifies conditions and suggests a model parameterisation for guaranteed high-quality-sample complexity improvement, and therefore can be used to guide the model architecture design and data collection procedure for finetuning large generative models particularly for usage in specialised domains such as healthcare, where collecting high-quality samples is difficult.

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

## A. Proof of Proposition 2

**Proposition 2** *Let $\mathcal{K} = \{1, \cdots, k\}$ denote a set of $k$ tokens. Let $\mathcal{Y}$ be the set of all finite length token sequences whose tokens all come from $\mathcal{K}$. Then $\mathcal{Y}$ has a one-to-one identification with the natural numbers. Let $\mathcal{P}_{\mathcal{Y}}$ be the set of probability mass functions over $\mathcal{Y}$, then the topological dimension $\dim(\mathcal{P}_{\mathcal{Y}}) = \infty$. If $\mathcal{Y}$ is instead the set of token sequences with length $\leq l$, then $\dim(\mathcal{P}_{\mathcal{Y}}) = O(k^l)$.*

*Proof of Proposition 2.* We can identify $\mathcal{Y}$ with the natural numbers as follows. Let $l$ denote the length of a token sequence. Since for each token in the sequence there are $k$ options, there are $k^l$ distinct sequences of length $l$.

We can define a bijective mapping from the set of $k^l$ sequences to the subset of natural numbers $\mathcal{S}^l := \left\{ \left( \sum_{i=1}^{l-1} k^i \right), \cdots, \left( \sum_{i=1}^{l} k^i \right) - 1 \right\}$ for $l \geq 2$ and $\mathcal{S}^1 := \{0, \cdots, k-1\}$. This is possible since the number of elements in $\mathcal{S}^l$ is $\left( \sum_{i=1}^{l} k^i \right) - 1 - \left( \sum_{i=1}^{l-1} k^i \right) + 1 = k^l$. Denote one such mapping $f_l$. Then define $f : \mathcal{Y} \to \mathbb{N}$:

$$f(y) = f_l(y) \text{ if length of } y = l \tag{19}$$

$f$ is well-defined because every $y$ has a unique length $l$. $f$ is invertible because $f_l$ is invertible for every $l$ and $\mathcal{S}^l \cap \mathcal{S}^{l'} = \emptyset$ and $\bigcup_{l=1}^{\infty} \mathcal{S}^l = \mathbb{N}$.

Thus, $\mathcal{P}_{\mathcal{Y}}$ is the set of probability mass functions whose sample space is $\cong \mathbb{N}$. That is, an element $P_Y \in \mathcal{P}_{\mathcal{Y}}$ is an infinite positive sequence which sums to 1.

Let $\Delta^d$ be the $d$-dimensional simplex. Note that $\mathcal{P}_{\mathcal{Y}} = \bigcup_{d=1}^{\infty} \Delta^d$, where $\Delta^d$ is viewed as a subset of $\Delta^{d+1}$ via the inclusion $\Delta^d \hookrightarrow \Delta^{d+1} : (p_1, \cdots, p_{d+1}) \mapsto (p_1, \cdots, p_{d+1}, 0, \cdots)$.

For each $d$, the topological dimension of the $d$-simplex $\Delta^d$ is $d$ [6]; this is to say, $d$ is the smallest number such that every open cover of $\Delta^d$ has an open refinement of order $d+1$. Therefore, the smallest number $n$ such that every open cover of $\bigcup_{d=1}^{D} \Delta^d$ has an open refinement of order $n+1$ is $D$. Hence, there is no finite $N$ such that the any open cover of $\mathcal{P}_{\mathcal{Y}} := \bigcup_{d=1}^{\infty} \Delta^d$ has an open refinement with order $N+1$. Therefore the topological dimension of $\mathcal{P}_{\mathcal{Y}}$ is $\infty$.

When the maximum sequence length is $l$ the cardinality of $\mathcal{Y}$ is finite and equal to $\sum_{i=1}^{l} k^i = \frac{k(1-k^l)}{1-k}$. $\mathcal{P}_{\mathcal{Y}}$ thus contains the set of positive sequences of length $\frac{k(1-k^l)}{1-k}$ which sum to 1, so $\mathcal{P}_{\mathcal{Y}}$ is a $\frac{k(1-k^l)}{1-k} - 1$-dimensional simplex, and therefore the topological dimension of $\mathcal{P}_{\mathcal{Y}}$ is $\frac{k(1-k^l)}{1-k} - 1 = O(k^l)$.

$\square$

## B. Proof of Theorem 3

**Lemma 1** *Under Condition 2, $\pi^{\dagger} \circ \tilde{\pi}^{-1}|_{\tilde{\pi}(\mathcal{X})}$ is a well-defined function.*

*Proof.*

**Inverse of non-injective functions.** In general, unless a function is *injective* [7], its inverse is not a function, but only a set map. For instance, since $\tilde{\pi}$ is many-to-one, so *not* injective, $\tilde{\pi}^{-1}$ would take an element from $\mathcal{P}_{\mathcal{Y}}$ and return a *subset* of $\mathcal{X}$, rather than a single element. Let $\tilde{\pi}^{-1}|_{\tilde{\pi}(\mathcal{X})}$ denote the inverse of $\tilde{\pi}^{-1}$ restricted to its image $\tilde{\pi}(\mathcal{X})$.

$\pi^{\dagger} \circ \tilde{\pi}^{-1}|_{\tilde{\pi}(\mathcal{X})}$ **is a well-defined function.** It follows that $\pi^{\dagger} \circ \tilde{\pi}^{-1}|_{\tilde{\pi}(\mathcal{X})}$ takes a point $P_Y$ in the image of $\tilde{\pi}$, $\tilde{\pi}(\mathcal{X})$, map it to its preimage $\tilde{\pi}^{-1}(P_Y)$, and map all points in the preimage $\tilde{\pi}^{-1}(P_Y)$ through $\pi^{\dagger}$ to $\pi^{\dagger}(\tilde{\pi}^{-1}(P_Y))$. For any two points $\boldsymbol{x}_1, \boldsymbol{x}_2 \in \tilde{\pi}^{-1}(P_Y)$, we have $\tilde{\pi}(\boldsymbol{x}_1) = \tilde{\pi}(\boldsymbol{x}_2)$, and then Condition 2 implies $\pi^{\dagger}(\boldsymbol{x}_1) = \pi^{\dagger}(\boldsymbol{x}_2)$. Therefore, for any $P_Y \in \mathcal{P}_{\mathcal{Y}}$, $\pi^{\dagger}(\tilde{\pi}^{-1}(P_Y))$ is a set containing exactly one element, so $\pi^{\dagger} \circ \tilde{\pi}^{-1}|_{\tilde{\pi}(\mathcal{X})}$ is a well-defined function. $\square$

**Theorem 3** *We work under Assumptions 2, 3, 4 and 1. For some $D$, there exists a Lipschitz invertible function $\tilde{\phi} : \mathcal{V} \to \tilde{\pi}(\mathcal{X})$, $\tilde{\Theta} \in \mathbb{R}^{N \times (D+1)}$ and $\tilde{\tau}^{\circ} : \mathcal{X} \to \Delta^D$ s.t. $\tilde{\pi} = \tilde{\phi} \circ \tilde{\Theta}\tilde{\tau}^{\circ}$, and there is a Lipschitz continuous function $\bar{\pi}^{\dagger} : \Delta^D \to \Delta^D$ s.t. $\pi^{\dagger} = \tilde{\phi} \circ \tilde{\Theta}\bar{\pi}^{\dagger} \circ \tilde{\tau}$.*

---

[6] Theorem 5, https://personal.colby.edu/ sataylor/teaching/F14/MA331/TopologicalDimension.pdf

[7] Injective essentially means one-to-one. Formally, a function $f$ is injective if $f(\boldsymbol{x}_1) \neq f(\boldsymbol{x}_2)$ whenever $\boldsymbol{x}_1 \neq \boldsymbol{x}_2$.

*Proof of Proposition 3.* **Step 1. Show that there exists $\tilde{\Theta}$, $\tilde{\tau}^\circ$ and Lipschitz $\tilde{\phi}$ such that $\tilde{\pi} = \tilde{\phi} \circ \tilde{\Theta}\tilde{\tau}^\circ$.**

By Condition 4, there is some invertible $L_\phi$-Lipschitz function $\tilde{\phi} : \mathcal{V} \to \tilde{\pi}(\mathcal{X})$, where $\mathcal{V}$ is some convex polygon. Therefore, there is a finite set $\mathcal{V}_{D+1} = \{v_d\}_{d=1}^{D+1}$ such that every $v \in \mathcal{V}$ can be expressed as $v = \sum_{d=1}^{D+1} p_d v_d$ for some $p \in \Delta^D$. Let $\tilde{\Theta} \in \mathbb{R}^{N \times (D+1)}$ be the matrix such that its $d$-th column, $\tilde{\Theta}_{:,d}$, is equal to $v_d$. Then every $v \in \mathcal{V}$ can be written as $v := \tilde{\Theta}p$ for some $p \in \Delta^D$.

Since $\tilde{\phi}^{-1} \circ \tilde{\pi}$ is a function $\mathcal{X} \to \mathcal{V}$, then for every $x$, $\tilde{\phi}^{-1} \circ \tilde{\pi}(x)$ is in $\mathcal{V}$. Therefore, there exists some $p_x \in \Delta^D$ such that

$$\tilde{\phi}^{-1} \circ \tilde{\pi}(x) = \tilde{\Theta}p_x \tag{20}$$

Let $\tilde{\tau}^\circ : \mathcal{X} \to \Delta^D$ be s.t.

$$\tilde{\tau}^\circ(x) = p_x \tag{21}$$

then

$$\tilde{\phi}^{-1} \circ \tilde{\pi}(x) = \tilde{\Theta}\tilde{\tau}^\circ(x) \tag{22}$$

$$\tilde{\pi}(x) = \tilde{\phi} \circ \tilde{\Theta}\tilde{\tau}^\circ(x) \tag{23}$$

**Step 2. Let $\tilde{\tau}(x) = \tilde{\Theta}\tilde{\tau}^\circ(x)$. We show that under the shared-level-sets assumption, $\pi^\dagger \circ \tilde{\tau}^{-1}|_{\tilde{\tau}(\mathcal{X})}$ is well-defined.** We have the following equalities:

$$\tilde{\pi}^{-1}|_{\tilde{\pi}(\mathcal{X})} = \left(\tilde{\phi} \circ \tilde{\tau}\right)^{-1}|_{\tilde{\pi}(\mathcal{X})} \tag{24}$$

$$= \tilde{\tau}^{-1}|_{\mathcal{V}} \circ \tilde{\phi}^{-1} \tag{25}$$

$$\tilde{\pi}^{-1}|_{\tilde{\pi}(\mathcal{X})} \circ \tilde{\phi} = \tilde{\tau}^{-1}|_{\mathcal{V}} \tag{26}$$

$$= \tilde{\tau}^{-1}|_{\tilde{\phi}^{-1}(\tilde{\pi}(\mathcal{X}))} \tag{27}$$

$$= \tilde{\tau}^{-1}|_{\tilde{\tau}(\mathcal{X})} \tag{28}$$

Therefore,

$$\tilde{\pi}^{-1}|_{\tilde{\pi}(\mathcal{X})} \circ \tilde{\phi} = \tilde{\tau}^{-1}|_{\tilde{\tau}(\mathcal{X})} \tag{29}$$

$$\pi^\dagger \circ \tilde{\pi}^{-1}|_{\tilde{\pi}(\mathcal{X})} \circ \tilde{\phi} = \pi^\dagger \circ \tilde{\tau}^{-1}|_{\tilde{\tau}(\mathcal{X})} \tag{30}$$

$$\tag{31}$$

By Condition 2 and Lemma 1, $\pi^\dagger \circ \tilde{\pi}^{-1}|_{\tilde{\pi}(\mathcal{X})}$ is well-defined. Since $\pi^\dagger \circ \tilde{\pi}^{-1}|_{\tilde{\pi}(\mathcal{X})}$ is well-defined, so is $\pi^\dagger \circ \tilde{\tau}^{-1}|_{\tilde{\tau}(\mathcal{X})}$.

**Step 3. Show that under Assumptions 3, 4 and 1, $\pi^\dagger$ can be decomposed as $\tilde{\phi} \circ \psi \circ \tilde{\tau}$ for some Lipschitz function $\psi : \tilde{\tau}(\mathcal{X}) \to \tilde{\phi}^{-1}\left(\pi^\dagger(\mathcal{X})\right)$.** Note that 1. $\tilde{\phi}$ is invertible restricted to its image, 2. by Condition 3 and 4 $\tilde{\phi}^{-1}$ is defined on the image of $\pi^\dagger$, and 3. $\pi^\dagger \circ \tilde{\tau}^{-1}|_{\tilde{\tau}(\mathcal{X})}$ is well-defined. Therefore, we can factor $\pi^\dagger$ as

$$\pi^\dagger = \tilde{\phi} \circ \tilde{\phi}^{-1} \circ \pi^\dagger \circ \tilde{\tau}^{-1}|_{\tilde{\tau}(\mathcal{X})} \circ \tilde{\tau} \tag{32}$$

Therefore, define:

$$\psi : \tilde{\tau}(\mathcal{X}) \to \tilde{\phi}^{-1}\left(\pi^\dagger(\mathcal{X})\right) \subset \tilde{\tau}(X) \tag{33}$$

$$\psi := \tilde{\phi}^{-1} \circ \pi^\dagger \circ \tilde{\tau}^{-1}|_{\tilde{\tau}(\mathcal{X})} \tag{34}$$

$$= \tilde{\phi}^{-1} \circ \pi^\dagger \circ \tilde{\pi}^{-1}|_{\tilde{\pi}(\mathcal{X})} \circ \tilde{\phi} \tag{35}$$

is a composition of Lipschitz functions (by Assumptions 4 and 1) so is Lipschitz.

**Step 4. Finally show the assertion, that $\pi^\dagger = \tilde{\phi} \circ \tilde{\Theta} \bar{\pi}^\dagger \circ \tilde{\tau}^\circ$ for some Lipschitz $\bar{\pi}^\dagger : \Delta^D \to \Delta^D$.** Substituting in $\tilde{\tau}(x) = \tilde{\Theta} \tilde{\tau}^\circ(x)$, we obtain:

$$\pi^\dagger(x) = \tilde{\phi} \circ \psi \circ \tilde{\Theta} \tilde{\tau}^\circ(x) \tag{36}$$

Let $\boldsymbol{p}_x = \tau^\circ(x) \in \Delta^D$. We want to show that there is a Lipschitz continuous function $\bar{\pi}^\dagger : \Delta^D \to \Delta^D$ such that

$$\psi\big(\tilde{\Theta} \boldsymbol{p}_x\big) = \tilde{\Theta} \bar{\pi}^\dagger\big(\boldsymbol{p}_x\big) \tag{37}$$

For a given $\boldsymbol{p}_x$, we can try to solve the linear system in terms of $\bar{\pi}^\dagger(\boldsymbol{p})$. We know that it must be an element of the set:

$$\tilde{\Theta}^+ \psi\big(\tilde{\Theta} \boldsymbol{p}_x\big) + \mathrm{Ker}\big(\tilde{\Theta}\big), \tag{38}$$

where $\tilde{\Theta}^+$ denotes the pseudoinverse.

Take the intersection between this set and $\Delta^D$; the intersection is non-empty because $\psi$ lands in $\mathcal{V}$. Now we describe a procedure to choose a point in this intersection that is Lipschitz continuous wrt $\boldsymbol{p}_x$: we let $\bar{\pi}^\dagger$ map $\boldsymbol{p}_x$ to the centroid of the intersection between $\Delta^D$ and $\tilde{\Theta}^+ \psi\big(\tilde{\Theta} \boldsymbol{p}_x\big) + \mathrm{Ker}\big(\tilde{\Theta}\big)$. The intersection is one of two convex sets, so it is convex; so the centroid lie in this set.

We now proceed to show that $\bar{\pi}^\dagger$ is Lipschitz continuous, in two steps. 1. First we show that the centroid of the intersection is a smooth function of the location of its vertices. 2. Then we show that the location of the vertices is a piecewise smooth function of $\boldsymbol{p}_x$.

**We show the centroid of the intersection is a generically smooth function of the location of its vertices.** Note that the intersection of $\Delta^D$ and $\tilde{\Theta} \psi\big(\tilde{\Theta} \boldsymbol{p}_x\big) + \mathrm{Ker}\big(\tilde{\Theta}\big)$ is a convex high-dimensional polyhedron, denote it $\mathcal{S}(\boldsymbol{p}_x)$.

The centroid of a convex high-dimensional polyhedron can be computed as follows: every convex polyhedron admits a triangulation. Let the triangulation of $\mathcal{S}(\boldsymbol{p}_x)$, denote it by $\mathrm{T}(\mathcal{S}(\boldsymbol{p}_x))$. For the $i$th simplex in $\mathrm{T}(\mathcal{S}(\boldsymbol{p}_x))$, take its vertices $\{\boldsymbol{v}_{i0}, \cdots, \boldsymbol{x}_{in}\}$ where $n$ is the dimension of the polyhedron. The centroid of the simplex is given by $\mathrm{C}(i) = \frac{\boldsymbol{v}_{i0} + \cdots + \boldsymbol{v}_{in}}{n+1}$, and the volume $\mathrm{Vol}(i)$ is given by $\frac{1}{n!}\left| \begin{pmatrix} \boldsymbol{v}_0 & \cdots & \boldsymbol{v}_n \\ 1 & \cdots & 1 \end{pmatrix} \right|$. The centroid of $\mathcal{S}(\boldsymbol{p}_x)$, denoted $\mathrm{C}(\mathcal{S}(\boldsymbol{p}_x))$ is given by $\sum_i \mathrm{C}(i)\,\mathrm{Vol}(i)$; since the determinant function is a polynomial in the matrix entries, this is a vector field where each entry is a polynomial. Moreover, every vertex in the triangulation but is not on $\mathcal{S}(\boldsymbol{p}_x)$ is in the interior of $\mathcal{S}(\boldsymbol{p}_x)$. There is some $\epsilon$ small enough such that we can draw an $\epsilon$-ball around each such vertex such that the closure of the balls are all disjoint and still lie in the polyhedron. So, perturb move each vertex to a point on the boundary of its ball; this gives a new triangulation, but the centroid is not changed. Since we can do it for any $\epsilon' \leq \epsilon$, $\mathrm{C}(\mathcal{S}(\boldsymbol{p}_x))$ is constant in the interior vertices. Therefore, $\mathrm{C}(\mathcal{S}(\boldsymbol{p}_x))$ is a polynomial of its vertices.

The limiting case is when two vertices overlap; in this case, at least one element in the triangulation will collapse onto a face, which has volume zero, so its centroid will not contribute to the calculation of $\mathrm{C}(\mathcal{S}(\boldsymbol{p}_x))$ through the formula. Therefore, the centroid of a polyhedron is a polynomial of its vertices, including when two or more vertices overlap.

**We show that the location of the vertices is a piecewise-smooth function of $\boldsymbol{p}_x$.** Note that the set of points in $\mathcal{S}(\boldsymbol{p}_x)$ is described as follows: Suppose $\dim(\mathcal{S}(\boldsymbol{p})) = J \leq D$, and choose an orthonormal basis in $\mathbb{R}^{D+1}$ whose span contains the direction vectors in $\mathcal{S}(\boldsymbol{p})$:

$$\boldsymbol{b}_1, \cdots, \boldsymbol{b}_J \tag{39}$$

Extend this to an orthonormal basis whose span contains $\Delta^D$:

$$\boldsymbol{b}_1, \cdots, \boldsymbol{b}_J, \boldsymbol{b}_{J+1}, \cdots, \boldsymbol{b}_D \tag{40}$$

And finally extend this to $\mathbb{R}^D$:

$$\boldsymbol{b}_1, \cdots, \boldsymbol{b}_J, \boldsymbol{b}_{J+1}, \cdots, \boldsymbol{b}_D, \boldsymbol{b}_{D+1} \tag{41}$$

So we can express

$$\mathcal{S}(\boldsymbol{p}_x) = \left\{ \boldsymbol{s} \in \mathbb{R}^{D+1} \;\middle|\; s_i \geq 0, \sum_i s_i = 1, \boldsymbol{A}\boldsymbol{s} = \boldsymbol{A}\Big(\tilde{\Theta}^+\psi(\tilde{\Theta}\boldsymbol{p})\Big) \right\}, \tag{42}$$

where $\boldsymbol{A} = \left(\boldsymbol{I}_{D+1} - \boldsymbol{B}_J\boldsymbol{B}_J^\top\right)$, $\boldsymbol{B}$ is the matrix whose rows are the $D+1$ basis vectors, and $\boldsymbol{B}_J$ is the one taking its first $J$ rows.

Note that $\boldsymbol{I}_{D+1} - \boldsymbol{B}_J\boldsymbol{B}_J^\top = \boldsymbol{B}_{-J}\boldsymbol{B}_{-J}^\top$ where $\boldsymbol{B}_{-J}$ is the matrix containing $\boldsymbol{b}_{J+1}, \cdots, \boldsymbol{b}_{D+1}$ as rows. But $\mathbf{1}$ is orthogonal to the row space of $\boldsymbol{B}_J^\top$, so it is contained in the row space of $\boldsymbol{B}_{-J}^\top$, and hence $\boldsymbol{B}_{-J}\boldsymbol{B}_{-J}^\top$. Therefore, we can remove $\sum_i s_i = 1$ from the set of conditions.

Therefore, the set of conditions contains $D+1-J$ linearly independent conditions and $D+1$ inequalities.

An extrema, i.e. a vertex, is the solution of $D+1$ linearly independent equations where all $D+1-J$ linearly independent equality constraints are included, together with $J$ equations from saturating the inequality constraints. Since $\boldsymbol{I}_{D+1} - \boldsymbol{B}_J\boldsymbol{B}_J^\top$ has rank $D+1-J$, there is a subset of $D+1-J$ rows, call the new matrix constructed from these rows $\bar{\boldsymbol{B}}_{D+1-J}$. Select $J$ vectors from the standard basis which are linearly independent of the rows of $\bar{\boldsymbol{B}}_{D+1-J}$, and stack them into an invertible matrix $\boldsymbol{C}$. Then any vertex is a solution of one such equation

$$\boldsymbol{v}^* = \boldsymbol{C}^{-1}\begin{pmatrix} \bar{\boldsymbol{B}}_{D+1-J}\Big(\tilde{\Theta}^+\psi\big(\tilde{\Theta}\boldsymbol{p}_x\big)\Big) \\ \mathbf{0} \end{pmatrix} \tag{43}$$

provided it still satisfies the remaining inequality constraints. Here it is clear that any $\boldsymbol{v}^*$ varies smoothly with $\boldsymbol{p}_x$. When all vertices of $\mathcal{S}(\boldsymbol{p}_x)$ are in the interior of a 1-dimensional face of $\Delta^D$, they vary locally smoothly with $\boldsymbol{p}_x$. Since the centroid of $\mathcal{S}(\boldsymbol{p}_x)$ varies smoothly with its vertices, whenever its vertices vary smoothly with $\boldsymbol{p}_x$, the centroid also vary locally smoothly with $\boldsymbol{p}_x$. The only non-differentiability happens when one vertex moves out of $\Delta^D$ and another moves in. But around those points of $\boldsymbol{p}_x$ the centroid is still continuous wrt $\boldsymbol{p}_x$, so $\bar{\pi}^\dagger$ is piecewise differentiable function on a compact domain, and therefore is Lipschitz continuous. $\qquad\square$

## C. Convergence rates proofs

**Lemma 4.** (Metric on policies through $r_\pi$) *Define* $r_\pi(\boldsymbol{x}, \boldsymbol{y}) = \beta \log \frac{\pi(\boldsymbol{y} \mid \boldsymbol{x})}{\pi_{ref}(\boldsymbol{y} \mid \boldsymbol{x})}$ *for some fixed constant* $\beta > 0$, *and let*

$$d_r(\pi, \pi') = \|r_\pi - r_{\pi'}\|_\infty \tag{44}$$

*Then* $d_r(\cdot, \cdot)$ *defines a metric over any set* $\Pi$ *of functions* $\mathcal{X} \times \mathcal{Y} \to [0,1]$ *s.t.* $\forall \boldsymbol{x} \sum_{\boldsymbol{y}} \pi(\boldsymbol{y} \mid \boldsymbol{x}) = 1$, *and satisfies* $|r_\pi(\boldsymbol{x}, \boldsymbol{y})| \leq C$ *over* $\mathcal{X}$ *and* $\mathcal{Y}$.

*Proof of Lemma 4.* $d_r$ is well-defined on $\Pi$ since for any $\pi, \pi' \in \Pi$, $d_r(\pi, \pi') \leq \|r_\pi\|_\infty + \|r_{\pi'}\|_\infty \leq 2C < \infty$.

We can verify that $d$ is a metric on $\Pi$. Clearly, symmetry and positivity holds, and $d(\pi, \pi') = 0 \iff \pi = \pi'$, so we just need to check triangle inequality. Fix $\pi'' \in \Pi$,

$$d_r(\pi, \pi'') + d_r(\pi', \pi'') = \|r_\pi - r_{\pi''}\|_\infty + \|r_{\pi'} - r_{\pi''}\|_\infty \tag{45}$$
$$\geq \|r_\pi - r_{\pi''} - r_{\pi'} + r_{\pi''}\|_\infty \tag{46}$$
$$= \|r_\pi - r_{\pi'}\|_\infty \tag{47}$$
$$= d_r(\pi, \pi') \tag{48}$$

$$\square$$

**Proposition 7** (Concentration bound). *Let* $G$ *be a measure on* $(X, Y_w, Y_l)$ *and for any* $\pi \in \Pi \subseteq \mathring{\Pi}$ *(Def. 1) let*

$$R_G(\pi) = \mathbb{E}_G\left[\log \sigma\left(\beta \log \frac{\pi(Y_w|X)}{\pi_{ref}(Y_w|X)} - \beta \log \frac{\pi(Y_l|X)}{\pi_{ref}(Y_l|X)}\right)\right] \tag{49}$$

*be the preference loss. Further, let $(X_i, Y_{w,i}, Y_{l,i})_{i=1}^n$ be i.i.d. samples from $G$, and let $\hat{G}_n$ denote the empirical measure given by the samples, then*

$$P\left(\sup_{\pi \in \Pi} |R_G(\pi) - R_{\hat{G}_n}(\pi)| \geq \epsilon\right) \leq 2 \inf_{\alpha \in (0,1)} \text{Cov}\left(\Pi, d_r(\cdot, \cdot), \frac{\alpha\epsilon}{4}\right) e^{-\frac{2(1-\alpha)^2 n \epsilon^2}{4C^2}} \tag{50}$$

*where $d_r(\pi, \pi') = \|r_\pi - r_{\pi'}\|_\infty$.*

*Proof of Proposition 7.* Adapted from Elesedy (2022).

Fix $\pi, \pi' \in \Pi$.

$$
\begin{aligned}
|R_G(\pi) - R_G(\pi')| &= \left| \mathbb{E}_G\left[ \log \sigma\left( \beta \log \frac{\pi(Y_w|X)}{\pi_{\text{ref}}(Y_w|X)} - \beta \log \frac{\pi(Y_l|X)}{\pi_{\text{ref}}(Y_l|X)} \right) \right. \right. \\
&\qquad \left. \left. - \log \sigma\left( \beta \log \frac{\pi'(Y_w|X)}{\pi_{\text{ref}}(Y_w|X)} - \beta \log \frac{\pi'(Y_l|X)}{\pi_{\text{ref}}(Y_l|X)} \right) \right] \right|
\end{aligned}
\tag{51}
$$

$$
\begin{aligned}
&\leq \mathbb{E}_G\left[ \left| \log \sigma\left( \beta \log \frac{\pi(Y_w|X)}{\pi_{\text{ref}}(Y_w|X)} - \beta \log \frac{\pi(Y_l|X)}{\pi_{\text{ref}}(Y_l|X)} \right) \right. \right. \\
&\qquad \left. \left. - \log \sigma\left( \beta \log \frac{\pi'(Y_w|X)}{\pi_{\text{ref}}(Y_w|X)} - \beta \log \frac{\pi'(Y_l|X)}{\pi_{\text{ref}}(Y_l|X)} \right) \right| \right]
\end{aligned}
\tag{52}
$$

When $\sigma$ is the sigmoid, $\log \sigma$ is concave, so the above is upper bounded:

$$
\begin{aligned}
&\leq \mathbb{E}_G\left[ \left| \left( \beta \log \frac{\pi(Y_w|X)}{\pi_{\text{ref}}(Y_w|X)} - \beta \log \frac{\pi(Y_l|X)}{\pi_{\text{ref}}(Y_l|X)} \right) \right. \right. \\
&\qquad \left. \left. - \left( \beta \log \frac{\pi'(Y_w|X)}{\pi_{\text{ref}}(Y_w|X)} - \beta \log \frac{\pi'(Y_l|X)}{\pi_{\text{ref}}(Y_l|X)} \right) \right| \right]
\end{aligned}
\tag{53}
$$

$$
\begin{aligned}
&\leq \mathbb{E}_G\left[ \left| \left( \beta \log \frac{\pi(Y_w|X)}{\pi_{\text{ref}}(Y_w|X)} - \beta \log \frac{\pi'(Y_w|X)}{\pi_{\text{ref}}(Y_w|X)} \right) \right. \right. \\
&\qquad \left. \left. - \left( \beta \log \frac{\pi(Y_l|X)}{\pi_{\text{ref}}(Y_l|X)} - \beta \log \frac{\pi'(Y_l|X)}{\pi_{\text{ref}}(Y_l|X)} \right) \right| \right]
\end{aligned}
\tag{54}
$$

$$\leq \mathbb{E}_G[|(r_\pi(X, Y_w) - r_{\pi'}(X, Y_w)) - (r_\pi(X, Y_l) - r_{\pi'}(X, Y_l))|] \tag{55}$$

$$\leq \mathbb{E}_G[|(r_\pi(X, Y_w) - r_{\pi'}(X, Y_w))|] + \mathbb{E}_G[|(r_\pi(X, Y_l) - r_{\pi'}(X, Y_l))|] \tag{56}$$

$$\leq 2\|r_\pi - r_{\pi'}\|_\infty \tag{57}$$

Now let $\hat{G}_n$ be the empirical measure of $(X, Y_w, Y_l)$ with $n$ samples. And define

$$L_{\hat{G}_n}(\pi) = R_{\hat{G}_n}(\pi) - R_G(\pi) \tag{58}$$

Then

$$\left| L_{\hat{G}_n}(\pi) - L_{\hat{G}_n}(\pi') \right| = \left| R_{\hat{G}_n}(\pi) - R_G(\pi) - R_{\hat{G}_n}(\pi') + R_G(\pi') \right| \tag{59}$$

$$\leq \left| R_{\hat{G}_n}(\pi) - R_{\hat{G}_n}(\pi') + R_G(\pi') - R_G(\pi) \right| \tag{60}$$

$$\leq \left| R_{\hat{G}_n}(\pi) - R_{\hat{G}_n}(\pi') \right| + |R_G(\pi') - R_G(\pi)| \tag{61}$$

$$\leq 4\|r_\pi - r_{\pi'}\|_\infty \tag{62}$$

So now let $\mathcal{K}$ be a $\kappa$-cover of $\Pi$ in $d_r(\cdot, \cdot)$. Define the sets $D(\pi_k) = \{\pi \in \Pi \,:\, d_r(\pi_k, \pi) \leq \kappa\}$. Then

$$P\left(\sup_{\pi \in \Pi}\left|L_{\hat{G}_n}(\pi)\right| \geq \epsilon\right) = P\left(\bigcup_{\pi_k \in \mathcal{K}}\left\{\sup_{\pi \in D(\pi_k)}\left|L_{\hat{G}_n}(\pi)\right| \geq \epsilon\right\}\right) \tag{63}$$

$$\leq \sum_{\pi_k \in \mathcal{K}} P\left(\sup_{\pi \in D(\pi_k)}\left|L_{\hat{G}_n}(\pi)\right| \geq \epsilon\right) \tag{64}$$

Set $\kappa = \frac{\alpha\epsilon}{4}$ for $0 < \alpha < 1$. Using the above, for any $\pi \in D(\pi_k)$ we have

$$\left|L_{\hat{G}_n}(\pi) - L_{\hat{G}_n}(\pi_k)\right| \leq 4\|r_\pi - r_{\pi_k}\|_\infty \tag{65}$$

$$= 4d_r(\pi_k, \pi) \tag{66}$$

$$\leq 4\kappa \tag{67}$$

By triangle inequality:

$$\left|L_{\hat{G}_n}(\pi)\right| - \left|L_{\hat{G}_n}(\pi_k)\right| \leq \left|L_{\hat{G}_n}(\pi) - L_{\hat{G}_n}(\pi_k)\right| \tag{68}$$

So

$$\left|L_{\hat{G}_n}(\pi)\right| \leq 4\kappa + \left|L_{\hat{G}_n}(\pi_k)\right| \left|L_{\hat{G}_n}(\pi)\right| \leq \alpha\epsilon + \left|L_{\hat{G}_n}(\pi_k)\right| \tag{69}$$

Since the probability of the supremum of over a cover is greater than $\epsilon$ implies that the upper bound over the cover is greater than $\epsilon$, we have that the probability of the latter is at least the probability of the former:

$$P\left(\sup_{\pi \in \Pi}\left|L_{\hat{G}_n}(\pi)\right| \geq \epsilon\right) \leq \sum_{\pi_k \in \mathcal{K}} P\left(\sup_{\pi \in D(\pi_k)}\left|L_{\hat{G}_n}(\pi)\right| \geq \epsilon\right) \tag{70}$$

$$\leq \sum_{\pi_k \in \mathcal{K}} P\left(\alpha\epsilon + \left|L_{\hat{G}_n}(\pi_k)\right| \geq \epsilon\right) \tag{71}$$

$$\leq \sum_{\pi_k \in \mathcal{K}} P\left(\left|L_{\hat{G}_n}(\pi_k)\right| \geq \epsilon(1 - \alpha)\right) \tag{72}$$

$$\tag{73}$$

Then Hoeffding's inequality gives

$$P\left(\sup_{\pi \in \Pi}\left|L_{\hat{G}_n}(\pi)\right| \geq \epsilon\right) \leq \sum_{\pi_k \in \mathcal{K}} P\left(\left|L_{\hat{G}_n}(\pi_k)\right| \geq \epsilon(1 - \alpha)\right) \tag{74}$$

$$\leq 2|\mathcal{K}| \exp\left(-\frac{2(1 - \alpha)^2 n\epsilon^2}{|2C|^2}\right) \tag{75}$$

$$\square$$

**Proposition 8** (Covering number in terms of that of domain and range)**.** *Fix* $\tilde{\phi}, \tilde{\Theta}, \tilde{\tau}^\circ$ *and* $L_{\bar{\pi}}$. *For* $\mathbf{p} \in \tilde{\tau}^\circ(\mathcal{X})$, *let*

$$\bar{\Pi}_{\mathbf{p}} := \left\{ g(\cdot_y) = \tilde{\phi}\left(\tilde{\Theta}\bar{\pi}(\mathbf{p})\right)[\cdot_y] \; \middle| \; \tilde{\phi} \circ \tilde{\Theta}\bar{\pi} \circ \tilde{\tau}^\circ(\cdot_x)[\cdot_y] \in \Pi\left(\tilde{\phi}, \tilde{\Theta}, \tilde{\tau}^\circ, L_{\bar{\pi}}\right) \right\} \subseteq \mathcal{P}_{\mathcal{Y}} \tag{76}$$

*and recall the metric on $\mathcal{P}_{\mathcal{Y}}$:*

$$d_{\mathcal{P}_{\mathcal{Y}}}(p, q) := \left\| \beta \log \frac{p(\cdot_y)}{q(\cdot_y)} \right\|_{\infty} \tag{77}$$

*Then for $\kappa, \delta > 0$, denoting as $\Delta_\delta^D$ the $\delta$-cover of $\Delta^D$ under metric $d_{\Delta^D}$:*

$$\text{Cov}\left( \Pi\left(\tilde{\phi}, \tilde{\Theta}, \tilde{\tau}^\circ, L_{\bar{\pi}}\right), d_r, 3\kappa + 3L_\phi \|\tilde{\Theta}\|_p L_{\bar{\pi}} \delta \right) \tag{78}$$

$$\leq \sup_{p' \in \Delta_\delta^D} \text{Cov}\left(\bar{\Pi}_{p'}, d_{\mathcal{P}_{\mathcal{Y}}}, \kappa\right)^{\text{Cov}\left(\Delta^D, d_\Delta, \delta\right)} \tag{79}$$

*Proof of Proposition 8.* We wish to find the covering number of $\Pi$ with the metric $d_r$.

**Take covers of the domain and range of $\bar{\pi}$.** Take a $\delta$-cover of $\tilde{\tau}^\circ(\mathcal{X})$ with metric $d_\Delta$, denote it $\Delta_\delta^D$. For every $p' \in \Delta_\delta^D$, take a $\kappa$-cover of $\bar{\Pi}_{p'}$ with metric $d_{\mathcal{P}_{\mathcal{Y}}}$. Denote it by $\bar{\Pi}_{p',\kappa}$.

**Construct a set of maps from the domain of $\bar{\pi}$ to the range of $\bar{\pi}$, via the covers.** Let $\bar{h}_{\Delta_\delta^D}$ be a map from $\Delta_\delta^D$ s.t. for every $p'$, $\bar{h}_{\Delta_\delta^D}(p') \in \bar{\Pi}_{p',\kappa}$. For every such $\bar{h}_{\Delta_\delta^D}$, extend it to $\bar{h}$, a function whose domain is $\Delta^D$ as follows: for $p \in \Delta^D$, let:

$$\bar{h}(p) = \begin{cases} \bar{h}_{\Delta_\delta^D}(p), & \text{if } p \in \Delta_\delta^D \\ \bar{h}_{\Delta_\delta^D}(p') \text{ where } p' \text{ is selected randomly from } A(p), & \text{if } p \notin \Delta_\delta^D \end{cases}, \tag{80}$$

$$A(p) = \left\{ p' \in \Delta_\delta^D \;\middle|\; d_{\Delta^D}(p, p') = \min_{p'' \in \Delta_\delta^D} d_{\Delta^d}(p, p'') \right\}. \tag{81}$$

**Construct a set of maps $\mathcal{X} \times \mathcal{Y} \to [0, 1]$, denoted by $\mathcal{H}_{\delta,\kappa}$. This set will be proved to cover $\Pi$.** For every $\bar{h}$, define $h : \mathcal{X} \times \mathcal{Y} \to [0, 1]$:

$$h(x, y) = \bar{h}(\tilde{\tau}^\circ(x))[y] \tag{82}$$

Let $\mathcal{H}_{\delta,\kappa}$ denote all such $h$. It can be checked that $\mathcal{H}_{\delta,\kappa} \subset \Pi$:

1. $\bar{h}(\tilde{\tau}^\circ(x)) \in \bar{\Pi}_{p'}$ for some $p'$ so is in $\mathcal{P}_{\mathcal{Y}}$; therefore, $\sum_y \bar{h}(\tilde{\tau}^\circ(x))[y] = 1$.

2. Since $p' \in \tilde{\tau}^\circ(\mathcal{X})$, take $x'$ so that $p' = \tilde{\tau}^\circ(x')$. Then wlog for every $x$, there is some $p'$ and $x'$ such that,

$$\bar{h}(\tilde{\tau}^\circ(x))[y] = \bar{h}(p')[y] \tag{83}$$
$$= \bar{h}_{\Delta_\delta^D}(p')[y] \tag{84}$$
$$= \tilde{\phi} \circ \tilde{\Theta}\bar{\pi}(p')[y] \tag{85}$$
$$= \tilde{\phi} \circ \tilde{\Theta}\bar{\pi}(\tilde{\tau}^\circ(x'))[y], \tag{86}$$

where the third equality holds since $\bar{h}_{\Delta_\delta^D}(p')[y] \in \bar{\Pi}_{p'}$. Therefore for every $x'$, $\left| \beta \log \frac{\bar{h}(\tilde{\tau}^\circ(x))[y]}{\pi_{ref}(y|x)} \right| \leq C$.

3. Now let us show that for some $\bar{\pi}_h$, $h = \tilde{\phi} \circ \tilde{\Theta}\bar{\pi}_h \circ \tilde{\tau}^\circ \in \mathring{\Pi}$. For every $p \in \tilde{\tau}^\circ(\mathcal{X})$, $\bar{h}(p) \in \bar{\Pi}_{p',\kappa}$ for some $p'$, i.e. there is some $\bar{\pi}_p$ s.t. $\bar{h}(p) = \tilde{\phi}\left(\tilde{\Theta}\bar{\pi}(p')\right)$. Construct $\bar{\pi}_h(p) := \bar{\pi}_p(p')$.

We need to show that $\mathcal{H}_{\delta,\kappa}$ covers $\Pi$ with metric $d_r$. To this end we need to show that for every $\pi \in \Pi$, there is a $h_\pi \in \mathcal{H}_{\delta,\kappa}$, such that $d_r(\pi, h_\pi) < \epsilon(\delta, \kappa)$ for some small value $\epsilon$ which depends monotonically on $\delta$ and $\kappa$, and $\epsilon(0, 0) = 0$.

So consider $d_r(\pi, h)$ for some $\pi \in \Pi$ and $h \in \mathcal{H}_{\delta,\kappa}$:

$$d_r(\pi, h) = \|r_\pi - r_h\|_\infty \tag{87}$$

$$= \sup_{x \in \mathcal{X}, y \in \mathcal{Y}} |r_\pi(x, y) - r_h(x, y)| \tag{88}$$

$$= \sup_{x \in \mathcal{X}, y \in \mathcal{Y}} \left| \beta \log \frac{\pi}{\pi_{\text{ref}}}(x, y) - \beta \log \frac{h}{\pi_{\text{ref}}}(x, y) \right| \tag{89}$$

$$= \sup_{x \in \mathcal{X}} \left\{ \sup_{y \in \mathcal{Y}} \left\{ \left| \beta \log \frac{\pi}{\pi_{\text{ref}}}(x, y) - \beta \log \frac{h}{\pi_{\text{ref}}}(x, y) \right| \right\} \right\} \tag{90}$$

$$= \sup_{x \in \mathcal{X}} \left\{ \sup_{y \in \mathcal{Y}} \{ |\beta \log \pi(x, y) - \beta \log h(x, y)| \} \right\} \tag{91}$$

So it is sufficient to show that for every $\pi$ there is some $h_\pi$ such that for every $x$,

$$\sup_{y \in \mathcal{Y}} \{ |\beta \log \pi(x, y) - \beta \log h_\pi(x, y)| \} \tag{92}$$

$$= \| \beta \log \pi(x, \cdot_y) - \beta \log h_\pi(x, \cdot_y) \|_\infty \tag{93}$$

$$\leq \epsilon(\delta, \kappa) \tag{94}$$

**Decompose the distance between a general $\pi \in \Pi$ and $h \in \mathcal{H}_{\delta,\kappa}$ for a fixed $x$. This helps us later choose the $h_\pi$ which makes the bound small enough.** Let $\pi \in \Pi$, $h \in \mathcal{H}_{\delta,\kappa}$, $x \in \mathcal{X}$ and $x' \in \mathcal{X}$ s.t. $\boldsymbol{p}' := \tilde{\tau}^\circ(x') \in A(\tilde{\tau}^\circ(x))$.

$$\| \beta \log \pi(x, \cdot_y) - \beta \log h(x, \cdot_y) \|_\infty \tag{95}$$

$$\leq \| \beta \log \pi(x, \cdot_y) - \beta \log \pi(x', \cdot_y) \|_\infty \tag{96}$$

$$+ \| \beta \log \pi(x', \cdot_y) - \beta \log h(x', \cdot_y) \|_\infty \tag{97}$$

$$+ \| \beta \log h(x', \cdot_y) - \beta \log h(x, \cdot_y) \|_\infty \tag{98}$$

**Consider** $\| \beta \log \pi(x, \cdot_y) - \beta \log \pi(x', \cdot_y) \|_\infty$**.** We can show this term is bounded by $L\delta$.

Let $x' \in \tilde{\tau}^{\circ,-1}(\boldsymbol{p}')$ be s.t.

$$\| \beta \log \pi(x, \cdot_y) - \beta \log \pi(x', \cdot_y) \|_\infty \tag{99}$$

$$\leq \left\| \beta \log \frac{\pi(x, \cdot_y)}{\pi(x', \cdot_y)} \right\|_\infty \tag{100}$$

$$\leq d_{\mathcal{P}_\mathcal{Y}}(\pi(x, \cdot_y), \pi(x', \cdot_y)) \tag{101}$$

$$\leq L_\phi \|\tilde{\Theta}\|_p L_{\bar\pi} d_\Delta(\tilde{\tau}^\circ(x), \tilde{\tau}^\circ(x')) \tag{102}$$

$$\leq L_\phi \|\tilde{\Theta}\|_p L_{\bar\pi} \delta \tag{103}$$

**Consider** $\| \beta \log \pi(x', \cdot_y) - \beta \log h(x', \cdot_y) \|_\infty$**; we show that we can choose $h_\pi$ to make this be upper bounded by $\kappa$.** Since $\pi \in \Pi$, it can be written as $\pi(\cdot_x, \cdot_y) = \tilde\phi \circ \tilde\Theta \bar\pi \circ \tilde{\tau}^\circ(\cdot_x)[\cdot_y]$ for some $\bar\pi : \Delta^D \to \Delta^D$.

Note that $\pi(x', \cdot_y) \in \bar\Pi_{\tilde\tau(x')}$. And since $\tilde{\tau}^\circ(x') \in \Delta_\delta^D$, there is some $\bar{h}_{\bar\pi} \in \bar\Pi_{\tilde{\tau}^\circ(x'),\kappa}$ s.t. $d_{\mathcal{P}_\mathcal{Y}}(\pi(x', \cdot_y), \bar{h}_{\bar\pi}(\tilde{\tau}^\circ(x'))) \leq \kappa$. But expanding this,

$$\kappa \geq d_{\mathcal{P}_\mathcal{Y}}(\pi(x', \cdot_y), \bar{h}_{\bar\pi}(\tilde{\tau}^\circ(x'))) \tag{104}$$

$$= \left\| \beta \log \frac{\pi(x', \cdot_y)}{\bar{h}_{\bar\pi}(\tilde{\tau}^\circ(x'))[\cdot_y]} \right\|_\infty \tag{105}$$

$$= \left\| \beta \log \pi(x', \cdot_y) - \bar{h}_{\bar\pi} \circ \tilde{\tau}^\circ(x', \cdot_y) \right\|_\infty \tag{106}$$

So choose $h_\pi := \bar{h}_{\bar\pi} \circ \tilde{\tau}^\circ$.

From now on replace $h$ by $h_\pi$.

**Consider** $\|\beta \log h_\pi(x', \cdot_y) - \beta \log h_\pi(x, \cdot_y)\|_\infty$. **We show that this is bounded above by** $2\kappa + 2L\delta$. Let $\boldsymbol{p}_1', \boldsymbol{p}_2' \in A(\tilde{\tau}^\circ(x))$, then let $x_i' \in \tilde{\tau}^{\circ,-1}(\boldsymbol{p}_i')$.

$$
\|\beta \log h_\pi(x_1', \cdot_y) - \beta \log h_\pi(x_2', \cdot_y)\|_\infty
$$

$$
\leq \|\beta \log h_\pi(x_1', \cdot_y) - \beta \log \pi(x_1', \cdot_y)\|_\infty +
$$
$$
\|\beta \log \pi(x_1', \cdot_y) - \beta \log \pi(x_2', \cdot_y)\|_\infty +
$$
$$
\|\beta \log \pi(x_2', \cdot_y) - \beta \log h_\pi(x_2', \cdot_y)\|_\infty \tag{107}
$$
$$
\leq \kappa + \|\beta \log \pi(x_1', \cdot_y) - \beta \log \pi(x_2', \cdot_y)\|_\infty + \kappa \tag{108}
$$
$$
= 2\kappa + \beta \left\| \log \frac{\pi(x_1', \cdot_y)}{\pi(x_2', \cdot_y)} \right\|_\infty \tag{109}
$$
$$
\leq 2\kappa + L_\phi \|\tilde{\Theta}\|_p L_{\bar{\pi}} d_\Delta(\tilde{\tau}^\circ(x_1), \tilde{\tau}^\circ(x_2)) \tag{110}
$$
$$
\leq 2\kappa + L_\phi \|\tilde{\Theta}\|_p L_{\bar{\pi}} d_\Delta(\tilde{\tau}^\circ(x_1), \tilde{\tau}^\circ(x)) + L_\phi \|\tilde{\Theta}\|_p L_{\bar{\pi}} d_\Delta(\tilde{\tau}^\circ(x), \tilde{\tau}^\circ(x_2)) \tag{111}
$$
$$
\leq 2\kappa + 2L_\phi \|\tilde{\Theta}\|_p L_{\bar{\pi}} \delta \tag{112}
$$

If $\tilde{\tau}(x) \notin \Delta_\delta^D$, $h(x) = \bar{h}(\tilde{\tau}^\circ(x)) = \bar{h}_{\Delta_\delta}(\boldsymbol{p}'') = h(x'')$ with some $\boldsymbol{p}'' \in A(\tilde{\tau}^\circ(x))$ and $\tilde{\tau}^\circ(x'') = \boldsymbol{p}''$, therefore

$$
\|\beta \log h_\pi(x', \cdot_y) - \beta \log h_\pi(x, \cdot_y)\|_\infty = \|\beta \log h_\pi(x', \cdot_y) - r_{h_\pi}(x'', \cdot_y)\|_\infty \tag{113}
$$
$$
\leq 2\kappa + 2L_\phi \|\tilde{\Theta}\|_p L_{\bar{\pi}} \delta \tag{114}
$$

Therefore,

$$
\|r_\pi(x, \cdot_y) - \beta \log h_\pi(x, \cdot_y)\|_\infty \leq 3\kappa + 3L_\phi \|\tilde{\Theta}\|_p L_{\bar{\pi}} \delta \tag{115}
$$

Since the upper bound is constant in $x$, we can conclude that $d_r(\pi, h_\pi) \leq 3\kappa + 3L_\phi \|\tilde{\Theta}\|_p L_{\bar{\pi}} \delta$. So $\mathcal{H}_{\delta,\kappa}$ covers $\Pi$ in $d_r$ with radius $3\kappa + 3L_\phi \|\tilde{\Theta}\|_p L_{\bar{\pi}} \delta$.

Therefore, we have that

$$
\text{Cov}\Big(\Pi, d_r, 3\kappa + 3L_\phi \|\tilde{\Theta}\|_p L_{\bar{\pi}} \delta\Big)
$$
$$
\leq |\mathcal{H}_{\delta,\kappa}| \tag{116}
$$
$$
\leq \prod_{\boldsymbol{p}' \in \Delta_\delta^D} |\bar{\Pi}_{\boldsymbol{p}',\kappa}| \tag{117}
$$
$$
\leq \sup_{\boldsymbol{p}' \in \Delta_\delta^D} \text{Cov}(\bar{\Pi}_{\boldsymbol{p}'}, d_{\mathcal{P}_\mathcal{Y}}, \kappa)^{\text{Cov}(\Delta_\delta^D, d_{\Delta^D}, \delta)} \tag{118}
$$

$$\square$$

### C.1. Proof of Theorem 5

**Theorem 5**(Bounding sample complexity in terms of dimension) *We remain in the set up of Proposition 8. The covering number of $\Pi$ is bounded above by a function of $D$:*

$$
\text{Cov}\Big(\Pi, d_r, 3\kappa + 3L_\phi \|\tilde{\Theta}\|_p L_{\bar{\pi}} \delta\Big)
$$
$$
\leq \left( \frac{2L_\phi \|\tilde{\Theta}\|_p \sqrt{D}}{\kappa} \right)^{D \left( \frac{2\sqrt{D}}{\delta} \right)^D} \tag{119}
$$

*Set $\kappa = \frac{\epsilon}{48}$, we need*

$$
n(\epsilon, \omega) = \Omega\left( \frac{D}{\epsilon^2} \left( \frac{96 L_\phi \|\tilde{\Theta}\|_p L_{\bar{\pi}} \sqrt{D}}{\epsilon} \right)^D \log\left( \frac{96 L_\phi \|\tilde{\Theta}\|_p \sqrt{D}}{\epsilon} \right) - \log \omega \right) \tag{120}
$$

*samples to generalise. That is, whenever $n' \geq n(\epsilon, \omega)$, we have*

$$P\left(\sup_{\pi \in \Pi} |R_G(\pi) - R_{\hat{G}_{n'}}(\pi)| \geq \epsilon\right) \leq \omega \tag{121}$$

*Proof of Theorem 5.* We will bound both $\text{Cov}\big(\bar{\Pi}_{\boldsymbol{p}'}, d_{\mathcal{P}_{\mathcal{Y}}}, \kappa\big)$ and $\text{Cov}\big(\Delta^D, d_{\Delta}, \delta\big)$ in terms of $D$.

First consider $\text{Cov}\big(\bar{\Pi}_{\boldsymbol{p}'}, d_{\mathcal{P}_{\mathcal{Y}}}, \kappa\big)$. Recall $\bar{\Pi}_{\boldsymbol{p}'}$:

$$\bar{\Pi}_{\boldsymbol{p}'} = \left\{ g(\cdot_y) = \tilde{\phi} \circ \tilde{\Theta} \bar{\pi}(\boldsymbol{p}')[\cdot_y] \,\middle|\, \tilde{\phi} \circ \tilde{\Theta} \bar{\pi} \circ \tilde{\tau}^\circ(\cdot_x)[\cdot_y] \in \Pi \right\} \tag{122}$$

Now we create a Lipschitz function such that the image is $\bar{\Pi}_{\boldsymbol{p}'}$: note that $\tilde{\phi} \circ \tilde{\Theta}$ is $L_\phi \|\tilde{\Theta}\|_p$-Lipschitz, where we recall that $L_\phi$ is the Lipschitz constant for $\tilde{\phi}$ and $\|\tilde{\Theta}\|_p$ is the operator-$p$-norm of $\tilde{\Theta}$ on $\Delta^D$. For a given $\boldsymbol{p}'$, let $K(\boldsymbol{p}') = \left\{ \bar{\pi}(\boldsymbol{p}') \,\middle|\, \tilde{\phi} \circ \tilde{\Theta} \bar{\pi} \circ \tilde{\tau}^\circ(\cdot_x)[\cdot_y] \in \Pi \right\} \subseteq \Delta^D$. Then $\tilde{\phi} \circ \tilde{\Theta} : K(\boldsymbol{p}') \to \bar{\Pi}_{\boldsymbol{p}'}$.

**Now use the covering number of $K(\boldsymbol{p}')$ to bound that of $\bar{\Pi}_{\boldsymbol{p}'}$.**

$$\text{Cov}\big(\bar{\Pi}_{\boldsymbol{p}'}, d_{\mathcal{P}_{\mathcal{Y}}}, \kappa\big) \leq \text{Cov}\left(K(\boldsymbol{p}'), d_{\Delta}, \frac{\kappa}{L_\phi \|\tilde{\Theta}\|_p}\right) \leq \text{Cov}\left(\Delta^D, d_{\Delta}, \frac{\kappa}{L_\phi \|\tilde{\Theta}\|_p}\right) \tag{123}$$

**Finally, since all vectors on $\Delta^D$ have bounded $p$-norm, we can bound, for some constant $E(p, D)$ depending on the norm:**

$$\text{Cov}\big(\Delta^D, d_{\Delta}, \kappa\big) \leq \left(\frac{2E(p, D)L_\phi \|\tilde{\Theta}\|_p \sqrt{D}}{\kappa}\right)^D \tag{124}$$

This gives us:

$$\text{Cov}\Big(\Pi, d_r, 3\kappa + 3L_\phi \|\tilde{\Theta}\|_p L_{\bar{\pi}} \delta\Big) \tag{125}$$

$$\leq \sup_{\boldsymbol{p}' \in \Delta_\delta^D} \text{Cov}\big(\bar{\Pi}_{\boldsymbol{p}'}, d_{\mathcal{P}_{\mathcal{Y}}}, \kappa\big)^{\text{Cov}\big(\Delta^D, d_{\Delta}, \delta\big)} \tag{126}$$

$$\leq \text{Cov}\left(\Delta^D, d_{\Delta}, \frac{\kappa}{L_\phi \|\tilde{\Theta}\|_p}\right)^{\text{Cov}\big(\Delta^D, d_{\Delta}, \delta\big)} \tag{127}$$

$$\leq \left(\frac{2E(p, D)L_\phi \|\tilde{\Theta}\|_p \sqrt{D}}{\kappa}\right)^{D\left(\frac{2E(p,D)\sqrt{D}}{\delta}\right)^D} \tag{128}$$

And let $\kappa = L_\phi \|\tilde{\Theta}\|_p (L_{\bar{\pi}}) \delta$.

Then the covering number bound becomes

$$\text{Cov}\Big(\Pi, d_r, 6L_\phi \|\tilde{\Theta}\|_p L_{\bar{\pi}} \delta\Big) \leq \text{Cov}\big(\Delta^D, d_{\Delta}, L_{\bar{\pi}} \delta\big)^{\text{Cov}\big(\Delta^D, d_{\Delta}, \delta\big)} \tag{129}$$

$$\leq \left(\frac{2E(p, D)\sqrt{D}}{L_{\bar{\pi}} \delta}\right)^{D\left(\frac{2E(p,D)\sqrt{D}}{\delta}\right)^D} \tag{130}$$

Recall that the generalisation error bound is

$$P\left(\sup_{\pi \in \Pi} |R_G(\pi) - R_{\hat{G}_n}(\pi)| \geq \epsilon\right) \leq 2 \inf_{\alpha \in (0,1)} \text{Cov}\Big(\Pi, d_r, \frac{\alpha\epsilon}{4}\Big) e^{-\frac{2(1-\alpha)^2 n\epsilon^2}{4C^2}} \tag{131}$$

For simplicity let $\alpha = \frac{1}{2}$. So, set

$$\frac{\epsilon}{8} = 6L_\phi \|\tilde{\Theta}\|_p L_{\bar{\pi}} \delta \tag{132}$$

So

$$\delta = \frac{\epsilon}{48 L_\phi \|\tilde{\Theta}\|_p L_{\bar{\pi}}}. \tag{133}$$

and

$$\kappa = \frac{L_\phi \|\tilde{\Theta}\|_p L_{\bar{\pi}} \epsilon}{48 L_\phi \|\tilde{\Theta}\|_p L_{\bar{\pi}}} \tag{134}$$

$$= \frac{\epsilon}{48} \tag{135}$$

We have

$$P\left( \sup_{\pi \in \Pi} |R_G(\pi) - R_{\hat{G}_n}(\pi)| \geq \epsilon \right) \tag{136}$$

$$\leq \text{Cov}(\Pi, d_r, \epsilon/8) e^{-\frac{n\epsilon^2}{8C^2}} \tag{137}$$

$$\leq e^{-\frac{n\epsilon^2}{8C^2}} \left( \frac{96 L_\phi \left\| \tilde{\Theta} \right\|_p E(p, D) \sqrt{D}}{\epsilon} \right)^{D \left( \frac{96 L_\phi \|\tilde{\Theta}\|_p L_{\bar{\pi}} E(p, D) \sqrt{D}}{\epsilon} \right)^D} \tag{138}$$

For simplices, $E[p, D] \leq 1$.

So let's say we want the probility upper bound to be $\omega$, then the number of samples $n$ we need to generalise is

$$\omega = e^{-\frac{n\epsilon^2}{8C^2}} \left( \frac{96 L_\phi \left\| \tilde{\Theta} \right\|_p \sqrt{D}}{\epsilon} \right)^{D \left( \frac{96 L_\phi \|\tilde{\Theta}\|_p L_{\bar{\pi}} \sqrt{D}}{\epsilon} \right)^D} \tag{139}$$

$$n = \Omega \left( \frac{D}{\epsilon^2} \left( \frac{96 L_\phi \|\tilde{\Theta}\|_p L_{\bar{\pi}} \sqrt{D}}{\epsilon} \right)^D \log \left( \frac{96 L_\phi \left\| \tilde{\Theta} \right\|_p \sqrt{D}}{\epsilon} \right) - \log \omega \right) \tag{140}$$

$\square$

### C.2. Proof of Theorem 6

**Theorem 6** (Bounding sample complexity of learning without proxy) *Let* $\mathring{\Pi}\left(L_\phi \|\tilde{\Theta}\|_p L_{\bar{\pi}}\right)$ *be the subset of* $\mathring{\Pi}$ *where* $\pi$ *is* $L_\phi \|\tilde{\Theta}\|_p L_{\bar{\pi}}$*-Lipschitz. Set* $\kappa = \frac{\epsilon}{24}$, *we need*

$$\Omega \left( \frac{D'}{\epsilon^2} \left( \frac{48 L_\phi \|\tilde{\Theta}\|_p L_{\bar{\pi}} E'(p, D') \sqrt{D'}}{\epsilon} \right)^{D'} \log \left( \frac{48 L_\phi \|\tilde{\Theta}\|_p L_{\bar{\pi}} E'(p, D') \sqrt{D'}}{\epsilon} \right) - \log \omega \right) \tag{141}$$

*samples to generalise, where* $D' \gg D$ *, and* $E'(p, D') \gg 1$. *That is, whenever* $n' \geq n(\epsilon, \omega)$, *we have*

$$P\left( \sup_{\pi \in \mathring{\Pi}\left(L_\phi \|\tilde{\Theta}\|_p L_{\bar{\pi}}\right)} |R_G(\pi) - R_{\hat{G}_{n'}}(\pi)| \geq \epsilon \right) \leq \delta \tag{142}$$

*Proof of Theorem 6.* For learning $\Pi\left(\tilde{\phi}, \tilde{\Theta}, \tilde{\tau}^{\circ}, L_{\pi}\right)$, the covering number bound is as in Eq 138.

For learning $\mathring{\Pi}$ with the same Lipschitz constant (i.e. $L_{\phi}\|\tilde{\Theta}\|_{p}L_{\bar{\pi}}$) as above but without proxy data, the covering number bound can be read off from Elesedy (2022). Since $\mathcal{X}$ is a discrete space, we use the $p$-norm-induced metric in the embedding space of $\mathcal{X}$; denote the embedding function $f$. Additionally denote the feasible subset in $\mathcal{P}_{\mathcal{Y}}$ by $\mathcal{P}$. Denote the hypothesis class as $\mathring{\Pi}\left(L_{\phi}\|\tilde{\Theta}\|_{p}L_{\bar{\pi}}\right)$ to mean the subset with smallest Lipschitz-constant in the argument,

$$\text{Cov}\left(\mathring{\Pi}\left(L_{\phi}\|\tilde{\Theta}\|_{p}L_{\bar{\pi}}\right), d_{r}, 2L_{\phi}\|\tilde{\Theta}\|_{p}L_{\bar{\pi}}\delta + \kappa\right) \tag{143}$$

$$= \text{Cov}(\mathcal{Y}, d_{\mathcal{P}_{\mathcal{Y}}}, \kappa)^{\text{Cov}(f(\mathcal{X}), d_{p}, \delta)} \tag{144}$$

$$\leq \left(\frac{2E'(p, D')L_{\phi}\|\tilde{\Theta}\|_{p}L_{\bar{\pi}}\sqrt{D'}}{\kappa}\right)^{D'\left(\frac{2E'(p,D')\sqrt{D'}}{\delta}\right)^{D'}} \tag{145}$$

Setting $\kappa = L_{\phi}\|\tilde{\Theta}\|_{p}L_{\bar{\pi}}\delta$,

$$\text{Cov}\left(\mathring{\Pi}\left(L_{\phi}\|\tilde{\Theta}\|_{p}L_{\bar{\pi}}\right), d_{r}, 2L_{\phi}\|\tilde{\Theta}\|_{p}L_{\bar{\pi}}\delta + \kappa\right) \tag{146}$$

$$\leq \left(\frac{2E'(p, D')\sqrt{D'}}{\delta}\right)^{D'\left(\frac{2E'(p,D')\sqrt{D'}}{\delta}\right)^{D'}} \tag{147}$$

then setting $\epsilon/8 = 3L_{\phi}\|\tilde{\Theta}\|_{p}L_{\bar{\pi}}\delta$ (i.e. setting $\kappa = \frac{\epsilon}{24}$)

$$P\left(\sup_{\pi \in \Pi}|R_{G}(\pi) - R_{\hat{G}_{n}}(\pi)| \geq \epsilon\right) \tag{148}$$

$$\leq \text{Cov}\left(\mathring{\Pi}\left(L_{\phi}\|\tilde{\Theta}\|_{p}L_{\bar{\pi}}\right), d_{r}, 2L_{\phi}\|\tilde{\Theta}\|_{p}L_{\bar{\pi}}\delta + \kappa\right)e^{-\frac{n\epsilon^2}{8C^2}} \tag{149}$$

$$\leq e^{-\frac{n\epsilon^2}{8C^2}}\left(\frac{48L_{\phi}\|\tilde{\Theta}\|_{p}L_{\bar{\pi}}E'(p, D')\sqrt{D'}}{\epsilon}\right)^{D'\left(\frac{48L_{\phi}\|\tilde{\Theta}\|_{p}L_{\bar{\pi}}E'(p,D')\sqrt{D'}}{\epsilon}\right)^{D'}} \tag{150}$$

A similar analysis show that we need

$$n = \Omega\left(\frac{D'}{\epsilon^2}\left(\frac{48L_{\phi}\|\tilde{\Theta}\|_{p}L_{\bar{\pi}}E'(p, D')\sqrt{D'}}{\epsilon}\right)^{D'}\log\left(\frac{48L_{\phi}\|\tilde{\Theta}\|_{p}L_{\bar{\pi}}E'(p, D')\sqrt{D'}}{\epsilon}\right) - \log\omega\right) \tag{151}$$

$\square$

# D. Experiments

## D.1. Tempered Reward

We include a simple experiment to demonstrate the efficacy of our method on a common real-world scenario - regularisation in learned rewards from tempered softmax. We set the environment as follows:

- $\mathcal{X} = \mathbb{R}^5, \mathcal{P}_{\mathcal{X}} = \mathcal{N}(\mathbf{0}, \mathbf{I}_5)$

- $\mathcal{Y} = 1, 2, 3$, so $\mathcal{P}_{\mathcal{Y}} = \Delta^2$ a two-simplex.

| | $\pi_{\text{ref}}$ | $\pi^\dagger$ | $\tilde{\pi}$ | $\tilde{\pi}_\theta$ | $\pi^\dagger{}_\theta$ |
|---|---|---|---|---|---|
| mean | 0.63 | 0.0 | 0.33 | 0.34 | **0.32** |
| std | 0.00 | 0.0 | 0.00 | 0.014 | 0.096 |

*Table 1.* Results for the Tempered Reward experiment

- $D = 1$.

- $\pi^\dagger : \mathbb{R}^5 \to \mathbb{R} \to \Delta^2$.

- $\log(\tilde{\pi}(y_k|x)) = \frac{\log(\pi^\dagger(y_k|x))}{T}$. Temperature $T = 5$.

- $\pi_{\text{ref}} = \text{Uniform}\{1, 2, 3\}$

The proxy policy and true policy are implemented as follows. $\tilde{\pi}$ is initialised as a neural network with two linear layers followed by an injective softmax layer; all weights and biases are sampled from $\text{Uniform}\left(-\frac{1}{\sqrt{\text{input features}}}, \frac{1}{\sqrt{\text{input features}}}\right)$:

$$\tilde{\pi} : x \mapsto \underbrace{\text{Linear}(\dim(\mathcal{X}), D) \mapsto \text{Linear}(D, |\mathcal{Y}| - 1)}_{\text{logits layer}} \mapsto \text{InjectiveSoftMax} \mapsto \tilde{\pi}(\cdot|x) \tag{152}$$

The true policy is initialised by scaling the logits layer of $\tilde{\pi}$ by $T$.

$$\pi^\dagger : x \mapsto \underbrace{\text{Linear}(\dim(\mathcal{X}), D) \mapsto \text{Linear}(D, |\mathcal{Y}| - 1) \mapsto T\cdot}_{\text{logits layer}} \mapsto \text{InjectiveSoftMax} \mapsto \pi^\dagger(\cdot|x) \tag{153}$$

The proxy policy model $\tilde{\pi}_\theta$ is parameterised as a concatenation of $\tilde{\tau}^\circ$, $\tilde{\Theta}$ and $\tilde{\phi}$, where the injectivity of $\tilde{\Theta}$ and $\tilde{\phi}$ are ensured by forcing $\tilde{\Theta}$ to be full-rank and parameterising $\tilde{\phi}$ with leaky-relu activation (which is injective) and full-rank linear layers. We parameterise the true policy model $\pi^\dagger{}_\theta$ using the trained $\tilde{\tau}^{\circ,*}$, $\tilde{\Theta}^*$ and $\tilde{\phi}^*$, together with $\bar{\pi}$ parameterised as a linear layer. The parameterisation of $\tilde{\pi}_\theta$ and $\pi^\dagger{}_\theta$.

$$\tilde{\pi}_\theta : x \mapsto \tilde{\phi}^*\left(\tilde{\Theta}^* \tilde{\tau}^{\circ,*}(x)\right) \tag{154}$$

$$\pi^\dagger{}_\theta : x \mapsto \tilde{\phi}^*\left(\tilde{\Theta}^* \bar{\pi}_\theta(\tilde{\tau}^{\circ,*}(x))\right) \tag{155}$$

We train the proxy policy model on 8000 proxy samples $\{(\tilde{x}_i, \tilde{y}_{w,i}, \tilde{y}_{l,i})\}_{i=1}^{8000}$ generated from $\tilde{\pi}$, then only finetune $\bar{\pi}$ from 35 true samples $\left\{\left(x_j^\dagger, y_{w,j}^\dagger, y_{l,j}^\dagger\right)\right\}_{j=1}^{35}$ generated from $\pi^\dagger$. We compare the KL divergences $\text{KL}(\pi^\dagger, \pi^\dagger{}_\theta)$ and $\text{KL}(\pi^\dagger, \tilde{\pi})$ to see if the learned $\pi^\dagger{}_\theta$ is robust against distribution shift $\pi^\dagger \mapsto \tilde{\pi}$. We repeat the experiments 6 times. The results are shown in Table 1.

### D.2. Empirical verification of Theorem 5 and Theorem 6

To help wider understanding, we attempt to empirically show the results of our theorem that the number of samples $n$ scales differently wrt the generalisation error $\epsilon$ for a model constructed with and without our proposed parameterisation. The setup is as follows, we have the prompt space $\mathcal{X} = \mathbb{R}^5$ and completion space $\mathcal{Y} = \{1, 2, 3, 4\}$, that is $\mathcal{P}_\mathcal{Y} = \Delta^3 \subset \mathbb{R}^4$; we set $D = 1$ again.

In this case, we construct the true and proxy policies to explicitly follow our parameterisation:

- The proxy policy $\tilde{\pi} : \mathcal{X} \xrightarrow{\tilde{\tau}^\circ} \Delta^D \xrightarrow{\tilde{\Theta}} \mathbb{R}^D \xrightarrow{\tilde{\phi}} \mathcal{P}_\mathcal{Y}$.

- So, by Theorem 3, the true policy is $\pi^\dagger : \mathcal{X} \xrightarrow{\tilde{\tau}^\circ} \Delta^D \xrightarrow{\bar{\pi}} \Delta^D \xrightarrow{\tilde{\Theta}} \mathbb{R}^D \xrightarrow{\tilde{\phi}} \mathcal{P}_\mathcal{Y}$.

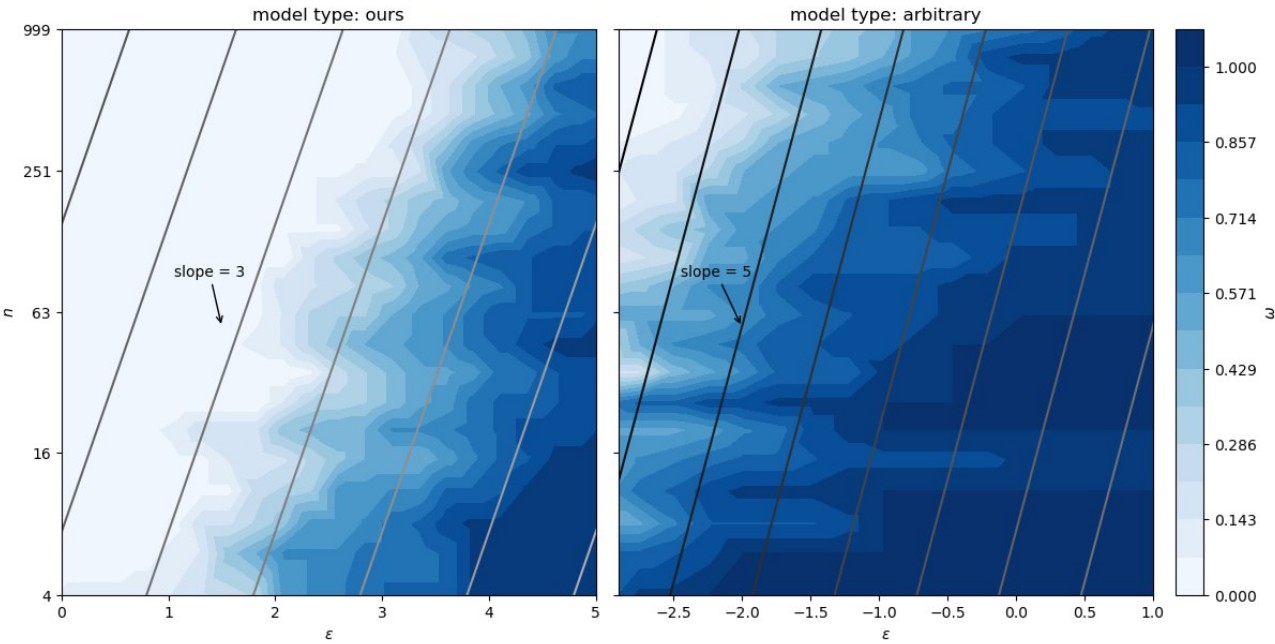

*Figure 3.* Log-log graph of number of samples $n$ against the maximum gap $\epsilon$.

The components $\tilde{\tau}^{\circ}$, $\tilde{\Theta}$, $\bar{\pi}$, $\tilde{\phi}$ are parameterised as neural networks such that the Lipschitz constants are all 1, in other words, $\|\tilde{\Theta}\|_2 = 1$, $L_{\tilde{\phi}} = 1$ and $L_{\bar{\pi}} = 1$.

Given this setup, our results show that, for a fixed $\omega$, $n = O(\epsilon^{-3})$ for models using our parameterisation and $n = O(\epsilon^{-5})$ for an arbitrary model. To verify this, we use the following policy models:

- For our parameterisation, we obtain samples $\{\hat{\pi}_i^{\dagger}\}_{i=1}$ from the hypothesis class of $\pi^{\dagger}$ by fixing $\tilde{\tau}^{\circ}$, $\tilde{\Theta}$, $\tilde{\phi}$ and sampling the different adapters $\hat{\bar{\pi}}$ from the class of 1-Lipschitz functions from $\Delta^1$ to $\Delta^1$.

- For an arbitrary model, we obtain samples $\{\hat{\pi}_j\}_j$ by using a general neural network parameterisation and sampling them from the class of 1-Lipschitz functions from $\mathcal{X}$ to $\mathcal{P_Y}$.

Although the actual bounds in Theorems 5 and 6 uses the supremum over $\hat{\pi}^{\dagger}$ or $\hat{\pi}$, to verify with supremum is computationally expensive, so we settle for only using the samples of $\hat{\pi}^{\dagger}$ and $\hat{\pi}$ directly.

Given all of this, we make a log-log graph with the number of samples $n$ and the maximum gap $\epsilon$ as axis and show the possible values of $\omega$ as color; the figure is shown in . As this is a log-log graph, our theoretical results says that the contour lines for a specific $\omega$ should have slope 3 and for the arbitrary parametrisation should have slope 5, we have ploted these lines in black and It can be seen from the plots that the contours are either in rough agreement with the plots, or the slope is slightly below the that of the plots, which is expected as the result is an upper bound.

