# OpenReview forum: "When Can Proxies Improve the Sample Complexity of Preference Learning?"
_ICML.cc/2025/Conference — ICML 2025 poster_

### Official Review · Reviewer_aNgX · 2025-03-10

**Overall Recommendation:** 4

**Summary:**

This paper discusses the sample complexity of optimizing LLMs with proxy rewards to improve the true policy. The authors give sufficient conditions, under which the proxy data is guaranteed to improve the sample complexity of learning the true policy. In general, I think this is an important topic as we are usually unsure of whether the data for fine-tuning LLMs is reliable enough to reflect the true objective, this paper gives us insight on how to address this important and practical problem.

**Claims And Evidence:**

Yes

**Essential References Not Discussed:**

I don't see any.

**Experimental Designs Or Analyses:**

NA

**Methods And Evaluation Criteria:**

NA

**Other Comments Or Suggestions:**

No

**Other Strengths And Weaknesses:**

See in questions.

**Questions For Authors:**

1.	I understand this is a paper focusing on theories.  But I am curious about the parameterization part in section 4.3. How are we able to constitute the three functions, i.e. the embedding function, the linear map and the injective function? Will they be possible to integrate into current architecture of LLMs? This may be crucial in transferring theory into practice.
2.	How will the level of distributional shift between proxy data and true data affect the results of this paper?
3.	Minors: writing issues, e.g. “iff” in condition 1

**Relation To Broader Scientific Literature:**

This paper discusses a practical problem in LLM fine-tuning that widely exist. Prior works tend to ignore the optimizad policy is actually a proxy policy.

**Theoretical Claims:**

Yes

---

> ### Author Rebuttal · Authors · 2025-04-01
>
> Thank you for acknowledging that our work addresses an important problem and for asking the interesting questions. We answer your questions below.
>
> **Questions For Authors:**
>
> **1. I understand this is a paper focusing on theories. But I am curious about the parameterization part in section 4.3. How are we able to constitute the three functions, i.e. the embedding function, the linear map and the injective function? Will they be possible to integrate into current architecture of LLMs? This may be crucial in transferring theory into practice.**
>
> This is an important question. Here is one example for how to instantiate each function.
>
> - Function (i): Softmax(NN(x)), where NN(x) is any arbitrary neural network, including fully connected networks or encoder architectures for language modelling such as attention mechanism.
> - Function (ii): A linear layer. This can be implemented by any weight matrix which maps from a lower to a higher-than-or-equal-to dimension, since a randomly generated matrix almost surely has full rank.
> - Function (iii). This can be composed of injective attention mechanisms, injective linear layers, injective fully-connected networks, and injective softmax layers. We explain them one-by-one: attention mechanisms can be implemented as injective by using the softmax attention, since applying softmax to a matrix will generically make it full rank (see e.g. [Bridging the Divide: Reconsidering Softmax and Linear Attention](https://arxiv.org/abs/2412.06590)  Han et al 2024). Injective linear layers as explained for function (ii) can be implemented as injective. Injective fully connected layers can be implemented by using a combination of full-rank weight matrices and injective activations such as leaky ReLU; see also references [Furuya et al., [Globally injective and bijective neural operators](https://proceedings.neurips.cc/paper_files/paper/2023/hash/b40d5797756800c97f3d525c2e4c8357-Abstract-Conference.html), 2023] and [Puthawala et al, [Globally injective relu networks](http://www.jmlr.org/papers/v23/21-0282.html), 2022]. Injective Softmax can be implemented as follows: x → [x, 0] → softmax([x,0]).
>
> **2. How will the level of distributional shift between proxy data and true data affect the results of this paper?**
>
> This is an interesting question! We just want to point our that distribution shift between proxy and true data can be large, so long as the conditions still hold.
>
> However, it is very interesting to consider the case where certain assumptions hold but others do not, or assumptions hold approximately. We have had a deeper look at our conditions and can relax them in two ways, please see our answer to reviewer 6yLU in ‘Strong conditions; hard-to-verify assumptions’.
>
> The in-between area of approximate but not full compliance with our conditions is indeed interesting to explore but require empirical evaluations, but as this paper focusses on theory, empirical evaluation is left outside the scope. We are currently working on extending the project to empirically test the sensitivity of the conditions in large-scale experiments.
>
> **3. Minors: writing issues, e.g. “iff” in condition 1**
>
> Please note that this is not a typo, iff is a mathematical shorthand for if and only if. However, we have since relaxed this condition, see our answer above.

---

### Official Review · Reviewer_tZK8 · 2025-03-11

**Overall Recommendation:** 3

**Summary:**

The paper studies the problem of reward hacking — eg training an agent when one only has access to proxy reward (and can maximize it) during training, but it leads to the true/gold reward going down. Concretely, the paper studies the setup where one has access to abundant proxy labels, but only few true/gold labels, and one wants to reduce the sample complexity of gold labels by using data with proxy labels.

This paper gives a theoretical analysis of this problem. It underpins 4 conditions between gold and proxy rewards that, if satisfied, can lead to possible sample complexity reduction. The paper then gives an algorithm to obtain such sample complexity reduction and proves convergence rates of their algorithm.

**Claims And Evidence:**

Yes, the claims are supported by proofs.

**Essential References Not Discussed:**

I am unaware of any such references.

**Experimental Designs Or Analyses:**

The paper has no experimental results.

**Methods And Evaluation Criteria:**

The paper is theoretical and has no experimental results.

**Other Comments Or Suggestions:**

None.

**Other Strengths And Weaknesses:**

None to note.

**Questions For Authors:**

Please look at the section **Theoretical Claims** for my questions/concerns.

**Relation To Broader Scientific Literature:**

The paper is related to reward hacking, as described in [2], [3].

**Theoretical Claims:**

> Our first condition says that two distinct prompts are mapped to the same response distribution under the true policy whenever they are under the proxy policy.

The above says “distinct prompts”, but in the stated condition 1, we do not enforce $x_1 \neq x_2$. This is nit-picking, not a serious flaw (since the case $x_1 = x_2$ is trivial if we assume the policy maps the same prompt to the same probability distribution deterministically).

# Conditions 1 - 4

I have tried to check the proofs in the paper (though not in absolute details), they look mostly okay to me. My main concern is the relevance of this paper’s theory in guiding practice. Could the authors provide one real use case/simulation to show why conditions 1-4 are reasonable? Eg I am uncertain if the Lipschitz continuity conditions make sense in the real world. Also, even experiments on a toy 1D bandit similar to would strengthen the paper a lot.

Another assumption that is probably hard to hold is the low dimensional encoding condition (condition 3). Could the authors give a real-world example of this?

**Disclaimer**: My research expertise is not theory, so if the other reviewers agree that the theory presented in this paper is relevant, than I would request the area chair to put more weight on those reviews.

# Offline model-based RL version

How does this paper’s theory hold for the more practical use case of offline model-based RL, which is the predominant use case of RLHF? More concretely, the most common use of RLHF is: (1) collect a lot of gold reward labels from humans, (2) train a reward model against these labels, which will be our proxy reward model, (3) train an agent against this proxy reward model.

This is the most common scenario where one studies reward hacking, to the best of my knowledge, as studied most prominently in [2], [3]. Could the authors extend the theory to this setup as well/could we make any useful predictions here?

How would the encoder/decoder setup look in real-world LLM cases?

# References

[1] Scaling Laws for Reward Model Overoptimization in Direct Alignment Algorithms, https://arxiv.org/abs/2406.02900

[2] Scaling Laws for Reward Model Overoptimization, https://arxiv.org/abs/2210.10760

[3] Reward Model Ensembles Help Mitigate Overoptimization, https://openreview.net/forum?id=dcjtMYkpXx

---

> ### Author Rebuttal · Authors · 2025-04-01
>
> Thank you for your insightful comments. We hope to address your main concerns below.
>
> **…says “distinct prompts”, but in the stated condition 1, we do not enforce x1≠x2…**
>
> Thanks for pointing this out, note that we have also relaxed the conditions 1 and 4:
>
> Condition 1 can be relaxed from iff (i.e. if and only if) to just if, with the following revised statement:
>
> Given $x_1, x_2 \in \mathcal X$ , we have: $\pi^\dagger(\cdot|x_1) = \pi^\dagger(\cdot | x_2) \text{ if } \tilde\pi(\cdot | x_1) = \tilde\pi(\cdot | x_2)$
>
> Moreover, Condition 4 can be shown to be equivalent to
>
> $d_{\mathcal{P_Y}}(\pi^\dagger(\cdot|x_1), \pi^\dagger(\cdot|x_2) ) \leq L d_{\mathcal{P_Y}}(\tilde\pi(\cdot|x_1), \tilde\pi(\cdot|x_2))$.
>
> Notice that this equivalent condition implies Condition 1. This means we can simplify our presentation and combine Conditions 1 and 4 to a single condition.
>
> Given this, perhaps the following version flows better:
>
> ‘Our first condition says that if two prompts are mapped to very different response distribution under the true policy then they cannot be mapped to very similar responses under the proxy policy.’
>
> **…relevance of this paper’s theory in guiding practice… if the Lipschitz continuity conditions make sense...**
>
> Using the medical example in the paper, (the revised) Condition $1$ (subsuming Conditions $1$ and $4$ from) means that if the gold/expert doctor thinks two symptoms are very different (i.e. they map to very different prescriptions), e.g., $|\pi^\dagger(\cdot|x_1) -  \pi^\dagger(\cdot|x_2)| = d$, then the proxy/student cannot believe the two symptoms are similar, at least there should be some constant $L$ such that the student should not think that the difference between two prescriptions is less than $d/L$ for any $d$. This essentially requires that the proxy policy has the correct (within a constant scaling factor) idea of ‘distance’ in the output (response distribution) space.
>
> In general any continuously differentiable function is Lipschitz continuous, what makes the difference is the constant $L$. Notice that we did not assume a fixed value for $L$ - it can be seen from the proof of Theorem $3$ that $L$ will impact the lipschitz constant of $\bar\pi$, which then by Theorem $5$ impact the sample complexity of learning with the proposed model parameterisation; this means that if the proxy data is ‘good’ at understanding the distance in the output space, then we need fewer samples to converge.
>
> We have also provided a simple experiment to illustrate a case when all four conditions are satisfied, please see our answer to reviewer pBxw for the description, under ‘Weakness 3’.
>
> **…toy 1D bandit..**
>
> We have provided a simple experiment. Please see our answer to reviewer pBxw for the description, under ‘Weakness 3’.
>
> **…the low dimensional encoding condition (condition 3)…**
>
> The low-dimensional encoding is similar to the standard low-dimensional manifold condition typically used in deep learning. For example, the last token embedding in a LLM represents, in a lower dimension, the next-token prediction distribution. The key difference here is that we assume the encoding to be bi-Lipschitz; note that this can be considered a mild condition since all invertible, continuously differentiable functions whose inverse is also continuously differentiable satisfy this condition. In most deep learning tasks, we assume that data can be fitted to a deep neural network, which are continuously differentiable, satisfying the condition; in a typical low-to-high-dimension decoder, the neural network can be implemented as an injective function, and its inverse is also typically continuously differentiable.
>
> **…offline model-based RL…most common use of RLHF is: (1) collect a lot of gold reward labels from humans, (2) train a reward model against these labels, which will be our proxy reward model, (3) train an agent against this proxy reward model.**
>
> This could be best explained in the context of our simple experiment: the experiment can be readily adapted to the RLHF pipeline you described: 1) collect gold or noisy reward labels from humans, 2) train a biased reward model **(biased since temperature is increased)**, (3) train a proxy policy from the biased reward model using our parameterisation (4) finetune on a small number of additional gold reward labels from humans.
>
> In general, our framework applies to contextual bandits, the subclass of RL problems relevant to current LLM training. The extension to full sequential RL is left outside the scope.
>
> **How would the encoder/decoder setup look in real-world LLM cases?**
>
> Due to space constraint, we have written the detailed explanation to this question in our response to **aNgX.** Please kindly refer there, thank you.

---

> > ### Comment · Reviewer_tZK8 · 2025-04-02
> >
> > The rebuttal has addressed most of my concerns, and I thank the authors for taking the time to write their rebuttal!
> >
> > **I have increased my score to 3**.

---

> > > ### Author Response · Authors · 2025-04-02
> > >
> > > Thank you very much for your review! If you have any further questions later, please feel free to let us know and we will be happy to address them!

---

### Official Review · Reviewer_pBxw · 2025-03-13

**Overall Recommendation:** 3

**Summary:**

The paper investigates how leveraging abundant proxy data---feedback from less (proxy) expert sources---can pre-train a model to obtain a low-dimensional representation, which then serves as a warm-start for learning a true policy from limited high-quality (true) expert data. Here, the true policy refers to the decision-making strategy that is directly aligned with the expert’s behavior, and it is the target in an imitation learning setting where the expert data are costly and scarce. Without incorporating proxy data, directly learning the true policy would require a much larger number of samples due to the high-dimensional nature of the task. The paper provides four key conditions on true expert policy and proxy-expert policy and a two-stage algorithm to allow improved sample efficiency.

The four conditions collectively ensure that the proxy data can be effectively adapted into the true policy: first, the true and proxy policies must share level sets (Condition 1), meaning that if two prompts yield the same output under one policy, they do so under the other; second, the output range of the true policy must be contained within that of the proxy policy (Condition 2), ensuring the proxy policy is sufficiently expressive; third, the proxy policy’s outputs must lie on a low-dimensional manifold (Condition 3), allowing them to be captured by a compact, finite-dimensional encoding; and finally, the transformation from proxy outputs to true policy outputs must be Lipschitz continuous (Condition 4), guaranteeing that small variations in the proxy representation result in only small changes in the true policy.

In the first stage of the two-stage learning algorithm, the model is trained on the abundant proxy data by decomposing the proxy policy into an encoder, a linear mapping, and a decoder, effectively capturing the essential low-dimensional structure. In the second stage, this pre-trained structure is fine-tuned using the limited expert data by learning a low-dimensional “adapter” function that corrects the proxy policy toward the true expert policy.

**Claims And Evidence:**

Claim: Pre-training on proxy data yields a low-dimensional representation that, when adapted, reduces the sample complexity of learning the true policy.

Evidence: Theorem 3 establishes the decomposition of the proxy policy into shared components and a low-dimensional adapter, while Theorems 5 and 6 provide sample complexity bounds that compare learning with and without proxy data.

**Essential References Not Discussed:**

Not aware of.

**Experimental Designs Or Analyses:**

No experiments.

**Methods And Evaluation Criteria:**

No algorithm evaluation was performed on benchmarks or datasets.

**Other Comments Or Suggestions:**

Line 116 (left col): What is the definition of sequence space $\ell^1$.

I believe $\ocircle$ is the composition operator, but defining it somewhere early might avoid confusion.

Equation 9: $\bar \pi^\dagger_\theta \rightarrow \pi^\dagger_\theta $  (let me know if I am wrong) ?

**Other Strengths And Weaknesses:**

Strengths:
I think the paper generally has nice presentation. There is a slight overload of notation, which makes parts of the paper hard to read, the broad claims and the different conditions are understandable, even for readers who are not deeply versed in theory.

Weakness 1:
The derived bounds appear to be vacuous at first glance. For instance, Theorem 5 seems to require a number of proxy data samples on the order of greater than $ \sqrt{D}^{D} $. For any reasonable value of $D$ (e.g., $D = 128$), this results in an astronomically large number, which raises concerns about the practical relevance of the bounds.

Weakness 2:
It is not entirely clear how the scale of $D’$ compares to $D$. The paper mentions that $D’ \gg D$, but given that the sample complexity is already extremely high with the current bounds, any further increase due to $D’$ seems to make the requirements even less feasible. This point might be a misunderstanding on my part, but it needs clarification.

Weakness 3:
Since the algorithm and theoretical results are intended for practical applications, it would be beneficial to include a toy example or a simple experiment. Such an example would help validate the conditions and demonstrate the practical improvement in sample complexity.

**Questions For Authors:**

I believe my questions are primarily mentioned in the weakness section.

**Relation To Broader Scientific Literature:**

The ideas in the paper are interesting because they relate to the widely followed paradigm of pre-training followed by fine-tuning, which is currently prevalent in large language models. The paper builds on this concept by showing, with a solid theoretical basis, how abundant proxy data can be used in pre-training to learn a low-dimensional representation, which is then fine-tuned using limited high-quality expert data.

**Theoretical Claims:**

I didn't check the proofs, as all of the proofs are in the appendix.

---

> ### Author Rebuttal · Authors · 2025-04-01
>
> Thank you for your encouraging comments and insightful questions. We hope to address your main concerns below.
>
> **Weakness 1 & 2: The derived bounds appear to be vacuous at first glance…**
>
> This is a good question. Note that in the number of samples bound in Theorem 5 and 6, there are three constants in front of $D$: the lipschitz constants $L_\phi$ and $L_{\bar\pi}$, and the matrix $p$-norm of $\|\tilde{\Theta}\|_p$; the lipschitz contant of a function intuitively sets its smoothness, and matrix norm can be seen as the lipschitz constant of the linear map given by the matrix. The **power** $D$ is applied the product of all terms inside the brackets (including $1/\epsilon$), and in particular if the product $= 1$ then raising to $D$ would remain it at $1$. Morally, with the same amount of training data, the higher the dimension of the input and output spaces of the hypothesis functions, the more we have to regularise the hypothesis space (i.e. the more smooth we need the functions) to achieve the same generalisation error. So here, the final bound is not necessarily an astronomically large number - it depends on how much we are willing to regularise (i.e. lower the constants of ) the hypothesis space.
>
> **Weakness 3: …a toy example or a simple experiment…**
>
> We include a simple experiment to demonstrate the efficacy of our method on a common real-world scenario - regularisation in learned rewards from tempered softmax. We set the environment as follows:
>
> - $\mathcal X = \mathbb R^5$
> - $\mathcal Y = \{1,2,3\}$, so $\mathcal{P_Y} = \Delta^2$ a two-simplex.
> - $D = 1$
> - $\pi^\dagger: \mathbb{R}^5 \to \mathbb R \to \Delta^2$
> - $\text{logit}(\tilde\pi(y_k|x))=\frac{\big(\log(\pi^\dagger(y_k|x))+\log(\sum_k\pi^\dagger(y_k|x))\big)}{T}$. Temperature $T=5$.
>
> We parametrise the policy model using our proposal, where in particular the final injective component is implemented using a combination of full-rank matrices and leaky-relu activation (which is injective); for more detailed discussion on how to implement the architecture please refer to our response to aNgX. We train a policy model using our proposed parameterisation, first on $8000$ proxy samples $(\tilde  x, \tilde   y_w, \tilde  y_l)$ generated from $\tilde{\pi}$, then only finetune $\bar\pi$ from $\pi^\dagger$ on $35$ true samples $(x^\dagger, y_w^\dagger, y_l^\dagger)$ generated from $\pi^\dagger$. We compare the KL divergences $KL(\pi^\dagger, \pi^\dagger_\theta)$ and $KL(\pi^\dagger, \tilde\pi)$ to see if the learned $\pi^\dagger_\theta$ is robust against distribution shift $\pi^\dagger \mapsto \tilde\pi$. We repeat the experiments $6$ times. The results are the following:
>
> |  | ref | true | proxy | pi_til | pi_dag |
> | --- | --- | --- | --- | --- | --- |
> | mean | 0.63 | 0.0 | 0.33 | 0.34 | 0.32 |
> | std | 0.00 | 0.0 | 0.00 | 0.014 | 0.096 |
>
> We see that the mean with $\pi^\dagger$(pi_dag) is closer to 0 than $\tilde\pi$(proxy), having learned on only 35 true samples.
>
> Due to the time constraints of the rebuttal, we were not able to fully tune the $\tilde\pi_\theta$ and $\pi^\dagger_\theta$ models. In the final version we will run larger experiments and include also comparisons with fully blackbox models for $\pi^\dagger$.
>
>
> **Line 116 (left col): What is the definition of sequence space ℓ1.**
>
> This is the space of (possibly infinite) sequences $\mathbf x$ such that $\sum_i^\infty |x_i| < \infty$. We will include this as a footnote in the paper for completeness.
>
> **I believe \ocircle is the composition operator, but defining it somewhere early might avoid confusion.**
>
> Yes it is, thanks for pointing it out. We will define it to help with clarity.
>
> **Equation 9: $\bar\pi^\dagger_\theta \to \pi^\dagger_\theta$ (let me know if I am wrong) ?**
>
> That is right! Thank you for the catch.

---

> > ### Comment · Reviewer_pBxw · 2025-04-03
> >
> > Thank you for the detailed clarifications and for including the small-scale experiment. I appreciate how you addressed my concerns about the theoretical bounds and the definitions, which certainly helps in understanding the framework better.
> >
> > At the same time, I believe it would be beneficial to include an experiment where all the assumptions and conditions are explicitly satisfied—perhaps by construction—to further validate the theoretical claims. Additionally, a more in-depth discussion around the theorems (particularly Theorems 5 and 6) in the paper would help in assessing the practical relevance of the derived bounds.
> >
> > Overall, I find the paper promising and valuable, but I remain inclined to maintain my current score, albeit with a note of low confidence in my assessment given that I might be missing certain aspects of the paper.

---

> > > ### Author Response · Authors · 2025-04-08
> > >
> > > Thank you for sharing your additional concerns. We would like to clarify that our previous toy example in which the proxy policy is a high temperature version of the true proxy satisfies the conditions set in our paper by construction and we can append a rigorous explanation for this in the main text.
> > >
> > > As we were reviewing our discussion on Theorems 5 and 6, we noticed that the critical definition of $R\_G$ and $R\_{\\hat G}$ were only present in the appendix; we apologise for this oversight and will move it to the main body in the final draft.
> > >
> > > Additionally, we would like to clarify that our results are of the form:
> > >
> > > $$
> > > P\\Big(\\sup\_{\\pi\\in\\Pi} |R\_{G}(\\pi) - R\_{\\hat{G}\_n}(\\pi)| \\geq \\epsilon\\Big) \\leq \\omega,
> > > $$
> > >
> > > where $R\_G(\\pi)$ denotes the true expected DPO loss and $R\_{\\hat{G}\_n}(\\pi)$ is the training loss for a random dataset of size $n$. Therefore, $|R\_{G}(\\pi) - R\_{\\hat{G}\_n}(\\pi)|$ is the generalisation gap, how much our empirical training loss is super-estimating the quality of the model. In other words, our results show how big the training dataset size $n$ needs to be such that, for the worst possible model, our training loss $R\_{\\hat{G}\_n}(\\pi)$ is super-estimating the true loss by $\\epsilon$ with probability at most $\\omega$. Logically then, when we train our model and minimise this loss, we know that true loss is also decreasing to maintain the $\\epsilon$  gap.
> > >
> > > To help wider understanding, we follow your request and attempt to empirically show the results of our theorem that $n$ scales differently wrt $\\epsilon$ for a model constructed with and without our proposed parameterisation. The setup is as follows, we have the prompt space $\\mathcal X = \\mathbb{R}^5$ and completion space $\\mathcal Y = \\{1,2,3,4\\}$, that is $\\mathcal{P\_Y} = \\Delta^3 \\subset \\mathbb{R}^4$. In this case, we construct the true and proxy policies to explicitly follow our parameterisation such that:
> > >
> > > - The proxy policy $\\tilde\\pi: \\mathcal X \\xrightarrow{\\tilde{\\tau^0}}\\Delta^1 \\xrightarrow{\\tilde{\\Theta}} \\mathbb{R}^2 \\xrightarrow{\\tilde\\phi} \\mathcal{P\_Y}$;
> > > - So, by Theorem 3, the true policy is $\\pi^\\dagger: \\mathcal X \\xrightarrow{\\tilde{\\tau^0}}\\Delta^1 \\xrightarrow{\\bar\\pi} \\Delta^1 \\xrightarrow{\\tilde{\\Theta}} \\mathbb{R}^2 \\xrightarrow{\\tilde\\phi} \\mathcal{P\_Y}$ .
> > >
> > > The components $\\tilde{\\tau^0}$, $\\tilde\\Theta$, $\\bar\\pi$, $\\tilde\\phi$ are parameterised as neural networks such that the Lipschitz constants are all 1, in other words, $\\|\\tilde\\Theta\\|\_2 = 1$*, $L\_{\\tilde\\phi} = 1$* and $L\_{\\bar\\pi}=1$.
> > >
> > > Given this setup, our results show that, for a fixed $\\omega$, $n = O(\\epsilon^{-3})$ for models using our parameterisation and $n = O(\\epsilon^{-5})$ for an arbitrary model. To verify this, we use the following policy models:
> > >
> > > - For our parameterisation, we obtain samples $\\{\\hat\\pi^\\dagger\_i\\}\_{i=1}$ from the hypothesis class of $\\pi^\\dagger$ by fixing $\\tilde{\\tau^0}$, $\\tilde\\Theta$, $\\tilde\\phi$ and sampling the different adapters $\\hat{\\bar\\pi}$ from the class of 1-Lipschitz functions from $\\Delta^1$ to $\\Delta^1$.
> > > - For an arbitrary model, we obtain samples $\\{\\hat\\pi\_j\\}\_j$ by using a general neural network parameterisation and sampling them from the class of 1-Lipschitz functions from $\\mathbb{R}^5$ to $\\mathcal{P\_Y}$.
> > >
> > > Although the actual bounds in Theorems 5 and 6 uses the supremum over $\\hat{\\pi}^\\dagger$ or $\\hat\\pi$, but to verify with supremum is computationally expensive, so we settle for only using the samples of $\\hat{\\pi}^\\dagger$ and $\\hat\\pi$ directly.
> > >
> > > Given all of this, we make a log-log graph with the number of samples $n$ and the maximum gap $\\epsilon$ as axis and show the possible values of $\\omega$ as color; the figure is shown in [this link](https://anonymous.4open.science/r/RewardHacking-ICML10939/epsilon\_omega.png). As this is a log-log graph, our theoretical results says that the contour lines for a specific $\\omega$ should have slope 3 and for the arbitrary parametrisation should have slope 5, we have ploted these lines in black and It can be seen from the plots that the contours are either in rough agreement with the plots, or the slope is slightly below the that of the plots, which is expected as the result is an upper bound.

---

### Official Review · Reviewer_6uLY · 2025-03-16

**Overall Recommendation:** 3

**Summary:**

The paper provides a theoretical framework for aligning policies under two different preference models—one “proxy” preference (e.g., from a reward model) and one “true” preference (e.g., from actual human judgments). The main contribution is a set of conditions under which the optimal policy derived from the proxy preference is (or is not) guaranteed to coincide with the optimal policy under the true preference. The authors present formal definitions, propose assumptions about how these preferences interact across prompts, and outline proofs showing how certain alignment guarantees hold when the stipulated conditions are satisfied. The paper’s conceptual focus is on identifying robust criteria for policy alignment, with theoretical arguments anchoring the central claims.

**Claims And Evidence:**

The submission claims that if certain conditions—particularly ones ensuring that both the proxy-optimal policy and the true-optimal policy produce the same or comparable outputs across prompts—are met, then the two policies essentially align. While the authors do offer formal theorems and illustrative examples, the practical relevance of these conditions is not entirely clear:

- Hard-to-verify assumptions:  The conditions are implicitly about the induced policy instead of the preference data distribution itself.  These  seem potentially unrealistic to check in real-world scenarios.
- Strong conditions: The conditions, especially the requirement in condition 1 that policies share the same level set across all prompts, seem very strong. It requires that for all prompts, as long as the proxy policy gives the same distribution for a pair of prompt, the same shall hold for the true distribution.
- Uncertain applicability to reward models vs. human preferences: Even if the theory holds in a simplified setting, it is unclear whether these conditions translate to real-life systems where the proxy preference model is learned from data while the true preference model is from human, which is hard to characterize in general.

Beyond these concerns, it would also help if the paper provided more empirical or real-world studies to demonstrate whether these theoretical conditions have any identifiable footprint in practical alignment tasks (e.g., under approximate or partial compliance).

**Essential References Not Discussed:**

N/A

**Experimental Designs Or Analyses:**

It would be nice if the authors could provide some discussions on how to verify the conditions empirically, especially when the reward is trained as a neural network from noisy human preference data, while the true preference is aggregated real human preference.

**Methods And Evaluation Criteria:**

The paper’s methods revolve around formal proofs and theoretical derivations rather than extensive empirical tests. While the authors propose evaluations based on comparing policies derived from proxy versus true preferences, this approach might not account for complexities in real-world settings, such as noisy data or imperfect model assumptions.

The paper might benefit from outlining a more concrete evaluation pipeline or set of benchmarks where one can empirically assess how well the proposed conditions correspond to measurable alignment in practice.

**Other Comments Or Suggestions:**

See above.

**Other Strengths And Weaknesses:**

See above.

**Questions For Authors:**

See above.

**Relation To Broader Scientific Literature:**

The paper contributes to ongoing discussions in the reinforcement learning and AI alignment communities about bridging the gap between a learned reward model (proxy preference) and the underlying human values or intentions (true preference).

**Theoretical Claims:**

The theoretical claims, especially the sample complexity analysis, look good to me.

---

> ### Author Rebuttal · Authors · 2025-04-01
>
> Thank you for your thoughtful comments, we hope to address your main concerns below.
>
> **Strong conditions; Hard-to-verify assumptions:**
>
> First we point out that while our conditions may be stringent, there exist a lot of ground truth -> proxy shifts that do meet our criteria; our results show that you can still get gains even under those shifts. For instance, increased temperature (from tempered softmax) is a common bias in learned (proxy) policy models; this prevalent form of bias satisfies our conditions 1-4.
>
> On the other hand, since the submission, we have managed to relax the conditions in the following ways:
>
> 1. Condition 1 can be relaxed from iff (i.e. if and only if) to just if, with the following revised statement:
>
>     Given $x_1, x_2 \in \mathcal X$ , we have: $\pi^\dagger(\cdot|x_1) = \pi^\dagger(\cdot | x_2) \text{ if } \tilde\pi(\cdot | x_1) = \tilde\pi(\cdot | x_2)$
>
>     Moreover, Condition 4 can be shown to be equivalent to
>
>     $d_{\mathcal{P_Y}}(\pi^\dagger(\cdot|x_1), \pi^\dagger(\cdot|x_2) ) \leq L d_{\mathcal{P_Y}}(\tilde\pi(\cdot|x_1), \tilde\pi(\cdot|x_2))$.
>
>     Notice that this equivalent condition implies Condition 1. This means we can simplify our presentation and combine Conditions 1 and 4 to a single condition about distributional distance, conducive to future work using distribution metrics to verify this condition.
>
> 2. Instead of requiring that (the revised) Condition 1 (which subsumes the original Conditions 1 and 4) holds for all $x_1, x_2 \in \mathcal X$, we only need to require them for $x_1, x_2$, $P_\mathcal{X}$-almost surely and all proofs go through. This helps future work verify the conditions as one can approximate with samples from $P_{\mathcal X}$.
>
> Additionally, as far as we are aware this is the first work that describes sufficient conditions on proxy preferences that allow them to be used to learn a (different) ground truth reward function. In general this problem is as hard as establishing guarantees for RLHF under distribution shifts. Given the prevalence of reward hacking and its impact (e.g., alignment faking [Greenblatt et al., [Alignment faking in large language models](https://arxiv.org/abs/2412.14093), 2024], rare behaviour [Jones et al., [Forecasting rare language model behaviors](https://arxiv.org/abs/2502.16797), 2025], sabotage [Benton et al., [Sabotage evaluations for frontier models](https://arxiv.org/abs/2410.21514), 2024]) we argue that steps towards a principled framework for proxy rewards is crucial for future LLM models.
>
> **Uncertain applicability to reward models vs. human preferences.**
>
> Thank you for this. One way to apply our conditions in the real world is as follows. Consider the following example: the true reward function can be described as some (unknown) linear combination of (unknown) functions $r^\dagger(\cdot) := g(f_1(\cdot), \ldots, f_n(\cdot))$ and the proxy reward function is an (unknown) combination a subset of these unknown functions $\tilde{r}(\cdot) := h(f_1(\cdot), \ldots, f_i(\cdot))$, where $i < n$ and $g$ may or may not be equal to $h$. Note that many of the reward hacking examples mentioned in the introduction can be described using this example:
>
> - **Oscillator performance (ground truth) and output amplitude & frequency (proxy)** [Bird & Layzell, 2002]: This proxy is missing factors that are needed to learn an oscillator of a specific frequency, as the proxy also rewards amplifiers that output noise. This resulted in learning a radio instead of an oscillator.
> - **Race ranking (ground truth) and player score (proxy)** [Clark & Amodei, 2016]: This proxy is missing key properties such as race finish time which led to agents that drove in circles to collect powerups, increasing their player score.
> - **Student success (ground truth) and accepting admissions offer (proxy)** [Golden, 2001]: The percentage of students accepting admissions offers is used by university ranking systems to indicate overall interest and selectivity. However, a university can maximize offer acceptance by rejecting highly-qualified applicants that they believe will attend another university, and so the proxy is clearly missing factors of student success.
>
> In cases such as these, Conditions 1 and 2 generically do not hold. This is because if functions are missing in the proxy reward then the ground truth and proxy policies cannot have the same level sets. There is also no guarantee that the image of the proxy policy contains the image of the ground truth policy. In general, our conditions can be used to invalidate potential proxies, given (incomplete) prior knowledge of the structure of true and proxy policies.
>
> **….more empirical or real-world studies….**
>
> While the time constraint of the rebuttal did not allow us to run real-world experiments, we provide a simple experiment addressing the case with tempered proxy policy, a common form of bias in practice. Please find it in our response to **pBxw** (addressing Weakness 3)

---

> > ### Comment · Reviewer_6uLY · 2025-04-05
> >
> > Thank you for your response. Some of my concerns are addressed and I have adjusted the score accordingly.

---

> > > ### Author Response · Authors · 2025-04-07
> > >
> > > Thank you for considering our additional comments, please let us know if there is anything else we could do to address more of your concerns!

---

### Decision · Program_Chairs · 2025-05-01

**Decision:**

Accept (poster)

**Comment:**

This paper studies the question of when maximizing a proxy reward is sufficient to learn the optimal policy. There appear to be a general consensus in favor of acceptance.

However, many aspects of the paper can be improved. For instance, as brought up by reviewer Reviewer 6uLY, the conditions defined in the paper are hard to verify and therefore it's unclear how useful the findings in this paper would be in practice.

Another drawback not brought up by reviewers is the missing of other related works. The effect of optimizing a proxy reward/model is studied widely in RL literature, including those concerning with poisoning attacks (reward is hacked by adversaries), sim-to-real transfer, model-based RL, etc. Sufficient conditions in similar flavors to those in this paper show up quite frequently in these works. A thorough literature review is needed to understand the connection and differences in these techniques.

I encourage the authors to improve the paper on these aspects for the camera ready version.